# Systematic characterization of chromodomain proteins reveals an H3K9me1/2 reader regulating aging in *C. elegans*

Xinhao Hou[1,3], Mingjing Xu[1,3], Chengming Zhu [1,3], Jianing Gao[1], Meili Li[1], Xiangyang Chen[1], Cheng Sun [1], Björn Nashan[1], Jianye Zang [1], Ying Zhou[1]✉, Shouhong Guang [1,2] ✉ & Xuezhu Feng[1]✉

The chromatin organization modifier domain (chromodomain) is an evolutionally conserved motif across eukaryotic species. The chromodomain mainly functions as a histone methyl-lysine reader to modulate gene expression, chromatin spatial conformation and genome stability. Mutations or aberrant expression of chromodomain proteins can result in cancer and other human diseases. Here, we systematically tag chromodomain proteins with green fluorescent protein (GFP) using CRISPR/Cas9 technology in *C. elegans*. By combining ChIP-seq analysis and imaging, we delineate a comprehensive expression and functional map of chromodomain proteins. We then conduct a candidate-based RNAi screening and identify factors that regulate the expression and subcellular localization of the chromodomain proteins. Specifically, we reveal an H3K9me1/2 reader, CEC-5, both by in vitro biochemistry and in vivo ChIP assays. MET-2, an H3K9me1/2 writer, is required for CEC-5 association with heterochromatin. Both MET-2 and CEC-5 are required for the normal lifespan of *C. elegans*. Furthermore, a forward genetic screening identifies a conserved Arginine124 of CEC-5's chromodomain, which is essential for CEC-5's association with chromatin and life span regulation. Thus, our work will serve as a reference to explore chromodomain functions and regulation in *C. elegans* and allow potential applications in aging-related human diseases.

The eukaryotic genome is packaged with histones and other proteins to form chromatin. Histones are subject to many types of post-translational modifications (PTMs), especially on their flexible tails. These modifications include acetylation and methylation of lysine (K) and arginine (R) and phosphorylation of serine (S) residues, and play fundamental roles in most biological processes that are involved in chromatin dynamics, gene expression regulation, and other DNA processes, such as repair, replication, and recombination. To interpret PTMs of histones, effectors/readers are recruited to provide a link between the chromatin landscape and functional outcomes[1–6].

[1]Department of Obstetrics and Gynecology, The First Affiliated Hospital of USTC, The USTC RNA Institute, Ministry of Education Key Laboratory for Membraneless Organelles & Cellular Dynamics, School of Life Sciences, Division of Life Sciences and Medicine, Biomedical Sciences and Health Laboratory of Anhui Province, University of Science and Technology of China, 230027 Hefei, Anhui, China. [2]CAS Center for Excellence in Molecular Cell Science, Chinese Academy of Sciences, 230027 Hefei, P. R. China. [3]These authors contributed equally: Xinhao Hou, Mingjing Xu, Chengming Zhu. ✉e-mail: caddiezy@ustc.edu.cn; sguang@ustc.edu.cn; fengxz@ustc.edu.cn

Over the decades, multiple families of conserved domains that recognize modified histones have been discovered. These domains include members of the structurally related "Royal family," such as chromodomain, tudor, PWWP and MBT (malignant brain tumor) repeat domains[7], which mainly recognize mono-, di- or trimethylated lysine residues. The chromodomain is an evolutionarily conserved region of approximately 30−60 amino acids[8]. It was first identified in polycomb (Pc) proteins and heterochromatin protein 1 (HP1)[3]. Despite the nature of recognition of methyl-lysine of histones[5,9], the biological functions of chromodomain proteins are highly diverse[3,5,9]. For example, the (Pc) proteins and HP1 play important roles in maintaining facultative and constitutive repressive heterochromatin, respectively, through their recognition of methyl-lysine residues on histone H3 (H3K27me and H3K9me)[3,5,10]. In contrast, CHD1 (chromo-ATPase/helicase DNA-binding protein 1) was implicated in transcriptionally active regions in chromosomes. Furthermore, the double chromodomains of CHD1 cooperate with each other to bind to methylated H3K4[5,11]. The spectrum of different chromodomain proteins may display a layer of regulation on the chromatin landscape, genome organization, and gene expression. Thus, systematically delineating the complicated functional network of chromodomain proteins may provide a further understanding of how and why these chromatin regulators function.

*C. elegans* has a number of beneficial features, making it a particularly powerful system for advancing our knowledge of chromodomain proteins and their functions in genome biology. The genome of *C. elegans* is compact, consisting of 100 Mb of DNA containing approximately 20,000 protein-coding genes. Importantly, due to the holocentric nature of *C. elegans* chromosomes, markers of heterochromatin, such as H3K9 methylation, are not concentrated at a single region on each chromosome. Instead, H3K9 methylation is enriched on chromosome arms in dispersed small domains[12]. This enables easy sequencing of repetitive sequences. Moreover, endogenous fluorescent tagging coupled with microscopy, functional omics and rapid genetic screening makes it possible to efficiently assess the biological function of chromodomain proteins. Furthermore, the well-studied development, aging, stress response and short life cycle of *C. elegans* enable us to investigate the diverse roles of chromodomain proteins in physiological processes.

The *C. elegans* genome encodes 21 proteins that contain chromodomains, 9 of which have identified homologs in humans, yet most of the chromodomain proteins remain poorly characterized. The two *C. elegans* HP1 homologs (HPL-1 and HPL-2) physically associate with transcriptional repressive heterochromatin. HPL-1 has been found in an LSD-1/CoREST-like complex (lysine-specific demethylase-1, corepressor for REST)[3,13]. HPL-2 is an H3K9me reader but associates with chromatin independent of H3K9me[14]. HPL-2 interacts with the zinc-finger protein LIN-13 and the H3K9me-binding MBT domain protein LIN-61, forming a complex that is part of the synthetic multivulva (synMuv) B group[15–18]. LET-418 is a well-characterized Mi-2 homolog. Strong loss-of-function alleles of *let-418* lead sterility and vulval defects, whereas temperature-sensitive *let-418* mutants are long-lived and stress resistant in a DAF-16-dependent manner[19]. LET-418 interacts with histone H3K4 demethylase SPR-5/LSD1 to maintain germline stem cell status[20]. CHD-3, another Mi-2 homolog, facilitates meiotic progression with LET-418 by ensuring genome stability[21]. CEC-1 and CEC-6 are two readers of H3K27me. CEC-1, together with the H3K9me reader CEC-3, contributes to the robust development, normal lifespan, and fitness of animals. CEC-3 and CEC-6 are required for germline immortality maintenance[22]. HERI-1 (also termed CEC-9) has been reported to antagonize nuclear RNAi by limiting H3K9me3 at siRNA-targeted genomic loci[23]. UAD-2 is a newly identified chromodomain protein that recognizes H3K27me3 and promotes piRNA focus formation and transcription[24]. It colocalizes with the upstream sequence transcription complex (USTC) at inner nuclear membrane (INM) foci[24,25]. MRG-1, an ortholog of human MORF4L2 (mortality factor 4 like 2), is also required for piRNA focus formation

and transcription[24]. In addition, MRG-1 interacts with the NURD complex, including MEP-1 and LET-418, and participates in piRNA-mediated gene silencing[26].

Susan M. Gasser and colleagues introduced a GFP reporter system using lacI/lacO repetitive arrays bearing the heterochromatic histone modifications H3K9me3 and H3K27me3 into *C. elegans*[27,28]. Genetic screens based on the subcellular localization of the GFP arrays identified regulators of heterochromatic formation and chromatin positioning at the nuclear periphery, including histone methyl transferases (HMTs) MET-2 and SET-25 and the chromodomain proteins CEC-4 and MRG-1[29–31]. In embryos, CEC-4 bound to the inner nuclear membrane and tethered heterochromatin through H3K9me to the nuclear periphery[30,32]. In intestinal cells, MRG-1 binds to euchromatin (H3K36me) and acts indirectly to anchor heterochromatin to the inner nuclear membrane[31].

Here, by systematically fluorescence tagging chromodomain proteins followed by ChIP-seq, genetic screening, and in vitro biochemical assays, we generated a resource of chromodomain proteins in *C. elegans*. The resource provides a comprehensive expression, regulation and functional map of the proteins in a eukaryotic system, which will not only serve as a reference to explore chromodomain proteins in *C. elegans* but also allow potential application in aging-related human diseases.

## Results

### A resource of fluorescence-tagged chromodomain proteins in *C. elegans*

The chromodomain is highly conserved in a wide range of organisms from *Schizosaccharomyces pombe*, *Drosophila melanogaster*, *Arabidopsis thaliana*, and *Caenorhabditis elegans* to *Mus musculus* and *Homo sapiens*. According to the presence of other types of domains, chromodomain proteins can be classified into 13 families[3,33]. These families include the chromodomain-helicase DNA-binding (CHD) family, the histone methyl transferase family, the HP1 family, the Polycomb family, the Msl-3 homolog family, the histone acetyltransferase (HAT) family, the retinoblastoma-binding protein 1 (RBBP1) family, the enoylCoA hydratase family, the SWI3 family, Ankyrin Family and the plant-specific chromomethylase family, etc.[33].

The *C. elegans* genome encodes 21 proteins that contain chromodomains, 9 of which have been reported to have homologs in humans (Supplementary Table 1). The homologs are divided into 4 conserved families, including the CHD family, HP1 family, histone acetyltransferase family, and Msl-3 homolog family. CHD family members contain paired tandem chromodomains and helicase domains. Among them, CHD-1 and CHD-7 are homologs of human CHD1 and CHD5, respectively. CHD-3 and LET-418 are two Mi-2 homologs. The HP1 family members HPL-1 and HPL-2 are also conserved. The mortality factor-related gene (MRG-1) is the ortholog of mammalian MRG15. The two histone acetyltransferases MYS-1 and MYS-2 are conserved in humans as well[33]. We performed sequence alignment and phylogenetic analyses of chromodomain proteins in *C. elegans* and *H. sapiens*. This result was largely consistent with previous work (Fig. 1a). Notably, we identified CEC-7 as a homolog of the Msl-3 family proteins of humans (Fig. 1a). Although most *C. elegans* chromodomain proteins (*cec* genes) have no clear homologs in humans, the chromodomains of these genes share high identity with the chromodomain of human HP1α (Supplementary Table 2).

To investigate the expression and function of these chromodomain proteins, we used CRISPR/Cas9 technology and constructed a number of fluorescently tagged transgenic strains by knocking in a *3xflag-gfp* tag in situ to the chromodomain genes. These strains were subjected to assays, including ChIP-seq, fluorescence microscopy, and physiological function annotation. By systematically analyzing the ChIP-seq datasets, we delineated the genome-wide distribution,

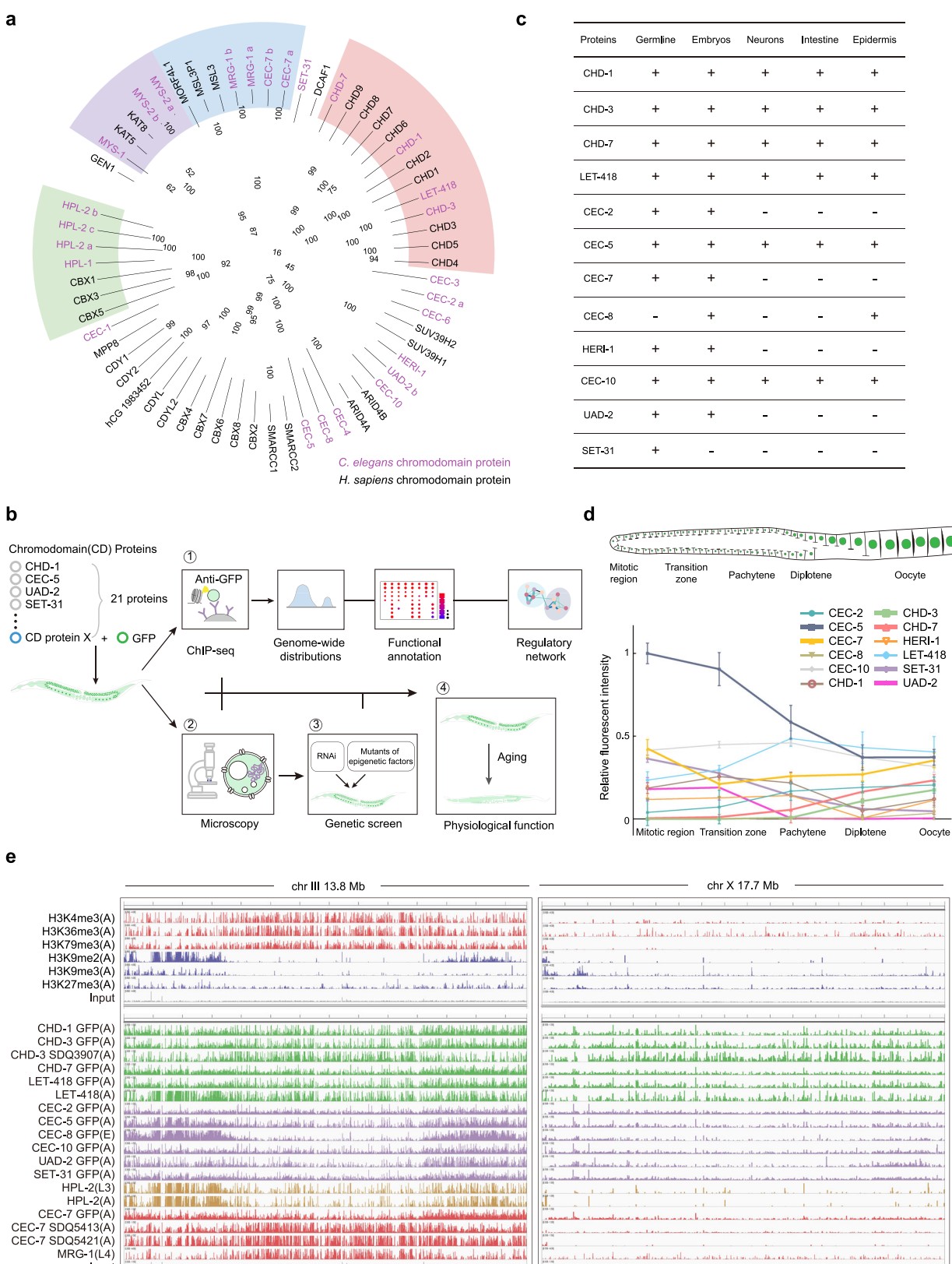

**c**

| Proteins | Germline | Embryos | Neurons | Intestine | Epidermis |
|---|---|---|---|---|---|
| CHD-1 | + | + | + | + | + |
| CHD-3 | + | + | + | + | + |
| CHD-7 | + | + | + | + | + |
| LET-418 | + | + | + | + | + |
| CEC-2 | + | + | - | - | - |
| CEC-5 | + | + | + | + | + |
| CEC-7 | + | + | - | - | - |
| CEC-8 | - | + | - | - | + |
| HERI-1 | + | + | - | - | - |
| CEC-10 | + | + | + | + | + |
| UAD-2 | + | + | - | - | - |
| SET-31 | + | - | - | - | - |

functional pathways, and epigenetic regulatory network in which these chromodomain proteins participate. In addition, we used the subcellular localization of each chromodomain protein as a reporter and screened for genetic factors that regulate the expression and localization of chromodomain proteins. The experimental pipeline is shown in Fig. 1b.

Of the 21 chromodomain genes in *C. elegans*, we successfully targeted 12 of them with a 3xFLAG-GFP fluorescent tag. Nearly all of the chromodomain proteins were broadly expressed in nuclei throughout germline, somatic cells, and embryos (Fig. 1c and Supplementary Figs. 1, 2a, b). Interestingly, the expression of many chromodomain proteins was dynamically altered in the germline (Fig. 1d and Supplementary

**Fig. 1 | Systematic analysis of chromodomain proteins in *C. elegans*.**
**a** Phylogenetic tree of chromodomain proteins in *C. elegans* (purple) and *H. sapiens* (black). Four conserved families are indicated by shadows in different colors, CHD family (red), HP1 family (green), histone acetyltransferase family (purple), and Msl-3 homolog family (blue). **b** Schematic diagram for systematic analysis of chromodomain proteins. See text for details. **c** Summary of the expression patterns of chromodomain proteins in the indicated cells. **d** Curve graph showing the expression profiles of chromodomain proteins in the germline. Mean ± SD; 4 germlines were used for calculation of each chromodomain protein. **e** Distribution of ChIP-seq signals of active (red), repressive (blue) chromatin marks (upper panel), and chromodomain proteins (lower panel) on chromosome III (left panel) and X (right panel). Chromodomain protein tracks are shown in different colors. CHD family (green), proteins with no clear homolog in human (purple), HP1 family (yellow), and Msl-3 homolog family (red). The development stage of animals for each sample is noted as E (embryos), L (larval), and A (adult). Source data are provided as a Source Data file.

Fig. 2a). For example, UAD-2 and SET-31 were expressed in the mitotic and meiotic regions but not in oocytes[24]. CEC-5 was highly expressed in the whole germline but declined during meiosis. The expression level of CHD-1 also reduced slightly during oogenesis. In contrast, CHD-3, CHD-7, and CEC-2 were not expressed in the mitotic and early meiotic regions but began to express in diplotene cells. The expression level of LET-418 increased in pachytene stage cells. Notably, CEC-7 exhibited a dramatic reduction in expression at the transition zone but was re-expressed in late pachytene cells (Fig. 1d, Supplementary Fig. 2a). CEC-8 was not expressed in the germline. The depletion of these chromodomain proteins slightly reduced the brood size (Supplementary Fig. 2c). Other chromodomain proteins were expressed constantly throughout the germline. Together, these data implied an orchestrated regulation of chromodomain proteins during germ cell maturation.

To identify genome binding signatures of the chromodomain proteins, we used a GFP antibody (#ab290) and conducted ChIP-seq experiments in adults or late embryos from the fluorescently tagged strains (Fig. 1e, Supplementary Figs. 6a–d, 7a, b and Supplementary Data 1). In addition, reported ChIP-seq datasets of several chromodomain proteins were downloaded from the NCBI GEO or modENCODE databases (Fig. 1e, Supplementary Figs. 6a–d, 7a, b and Supplementary Data 1)[34–36]. The quality of all ChIP-seq datasets was evaluated (Supplementary Figs. 3, 4a, b, 5a–f, Supplementary Table 3 and Supplementary Data 2). *C. elegans* chromosomes are organized into broad domains that differentiate the center of the chromosomes from the arms: active chromatin marks such as H3K4me3 and H3K36me3 have a similar distribution from centers to arms, while repressive histone marks, especially H3K9me1/2/3, are enriched at the distal chromosome arms (Supplementary Table 4, see "Methods")[12]. We mapped the binding locations of chromodomain proteins and compared the patterns to each other and to those of H3K9me2, H3K9me3, H3K27me3, H3K4me3, and H3K36me3 marks. Remarkably, approximately 44% of genomic regions are localized on chromosome arms. CHD-7 (58% on arms), CEC-5 (66% on arms), CEC-8 (97% on arms), CEC-10 (59% on arms), UAD-2 (77% on arms), HPL-2 (71% on arms in young adult; 68% on arms in L3), and SET-31 (62% on arms) were preferentially enriched at chromosome arms of autosomes. CHD-3 (35% on arms), CHD-1 (45% on arms), LET-418 (43% on arms in this study), CEC-2 (36% on arms), MRG-1 (32% on arms), and CEC-7 (46% on arms) were uniformly distributed from chromosome arms to centers (Fig. 1e and Supplementary Fig. 7a–c). The diverse genome distribution patterns implied distinct functions of chromodomain proteins in chromatin regulation.

## Association maps between chromodomain proteins and histone modifications on chromosome arms and centers

Chromodomain proteins typically bind specific histone modifications[3,5]. For example, CEC-4 recognizes H3K9me1/2/3, whereas MRG-1 may associate with H3K36me2/3 marks[31]. To determine the histone modifications bound by each chromodomain protein in vivo, we used three strategies and analyzed the association of chromodomain protein distribution signatures with H3K9me2/3, H3K27me3, H3K4me3, H3K36me3 and H3K79me3 marks on chromosome arms and centers. 1. We identified significant overlapped peaks of chromodomain proteins and histone modifications by using IntervalStats software package with a threshold of $P < 0.05$ (Fig. 2a, Supplementary Fig. 8a–c and Supplementary Table 5). 2. We plotted heatmaps and clustered peaks of each chromodomain protein by coverage of histone modifications (Fig. 2b–e, Supplementary Figs. 8d–g, and 9a). 3. We calculated the Pearson correlation coefficient of each pair of chromodomain protein and histone modification (Supplementary Figs. 10 and 11).

Chromodomain proteins displayed diverse propensities for histone modifications on chromosome arms that were enriched for both heterochromatin and euchromatin (Fig. 2a–f, Supplementary Figs. 8a, 8d–g, and 10, Supplementary Tables 5, 6). Strikingly, CEC-8, HPL-2, and CEC-5 were prominently enriched in H3K9me2 abundant regions (Fig. 2a–b, f, Supplementary Figs. 8d and 10, Supplementary Tables 5, 6). A small portion of CEC-5 targets were coated with H3K4me3 (Fig. 2b, f and Supplementary Table 6) and CEC-8 was moderately correlated with H3K79me3 (Fig. 2f and Supplementary Fig. 10, Supplementary Table 6). UAD-2, CHD-7, and SET-31 exhibited similar distribution patterns and preferentially associated with H3K27me3 and H3K79me3 marks (the association of UAD-2 with H3K79me3 was weak), yet weak signals of H3K9me2/3, H3K4me3, and H3K36me3 were also identified on these target sites (Fig. 2a, c, f and Supplementary Fig. 10, Supplementary Tables 5, 6). In contrast, CHD-3 (prominent), CHD-1 (detectable) and LET-418 (detectable) were correlated with H3K4me3 marks, whereas LET-418 was also related to H3K9me2 (Fig. 2a, d, f, Supplementary Figs. 8f and 10, Supplementary Tables 5, 6). MRG-1 and CEC-7 were mainly correlated with H3K36me3 and H3K79me3. MRG-1 also associated with H3K4me3, and CEC-7 might also associate with H3K9me2 and H3K4me3 (Fig. 2a, e, f, Supplementary Figs. 8g and 10, Supplementary Tables 5, 6).

In chromosome centers, which were enriched for active chromatin, most of the chromodomain protein targets, except CEC-8, were marked with H3K4me3 (Supplementary Figs. 8a, 9a, b, and 11, Supplementary Tables 5, 6). CEC-8 targets were specifically covered by H3K9me2 and HPL-2 also associated with H3K9me2 (Supplementary Figs. 8a, 9a, b, and 11, Supplementary Tables 5, 6). MRG-1 and CEC-7 correlated with H3K36me3 and H3K79me3 at the centers (Supplementary Figs. 8a, 9a, b, and 11, Supplementary Tables 5, 6). In addition, UAD-2, CHD-7, and SET-31 were also associated with H3K27me3, H3K36me3 and H3K79me3 (Supplementary Figs. 8a, 9a, b, and 11, Supplementary Tables 5, 6).

In addition, we analyzed the chromatin state of protein-coding genes and repetitive elements showing a certain combination of chromodomain proteins. The result was largely consistent with the "graded histone modification occupied" analysis (Fig. 2f and Supplementary Fig. 9b). For example, the "graded histone modification occupied" analysis showed CEC-5 GFP(A), CEC-8 GFP(E), HPL-2(L3), and HPL-2(A) targets were all associated with H3K9me2 (Fig. 2f and Supplementary Fig. 9b). A combination of CEC-5 GFP(A), CEC-8 GFP(E), HPL-2(L3), HPL-2(A) targets (both genes and repeats) were occupied by H3K9me2 as well (Supplementary Fig. 12a, b). CHD-1 GFP(A), CHD-3 GFP(A), and CHD-3 SDQ3907(A) targets were all associated with H3K4me3 (Fig. 2f and Supplementary Fig. 9b). A combination of CHD-1 GFP(A), CHD-3 GFP(A), and CHD-3 SDQ3907(A) targets (both genes and repeats) were also occupied by H3K4me3, although weaker H3K36me3 and H3K79me3 signals were also detected (Supplementary Fig. 12a, b).

Most chromodomain proteins were enriched in both euchromatin and heterochromatin, although our results may only reflect chromodomain protein occupancies and chromatin states of bulk tissues (Fig. 2f and Supplementary Fig. 9b). Interestingly, chromodomain

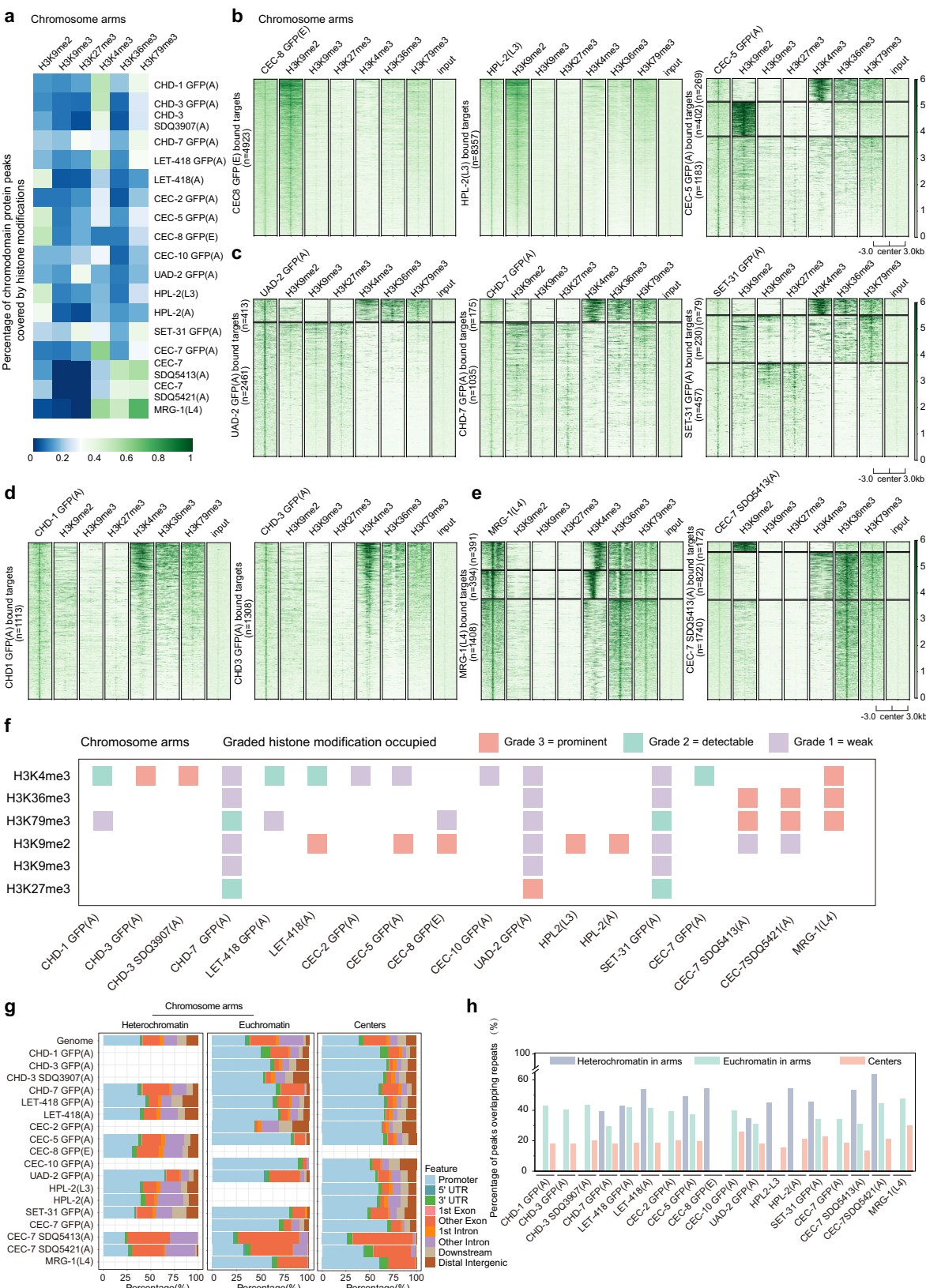

proteins exhibited different binding patterns in the two chromatin regions (Fig. 2g). In euchromatin, chromodomain proteins mainly bound to gene promoters, whereas in heterochromatin, chromodomain proteins targeted a large number of exons and introns (Fig. 2g). Consistently, most chromodomain proteins were enriched at promoters of actively transcribed genes, while mainly bound gene bodies

of silent genes (Supplementary Fig. 13a, b). In addition, chromodomain proteins showed a greater propensity for repetitive sequences in heterochromatin regions (Fig. 2h).

Collectively, we revealed distinct chromodomain protein-histone modification association maps in chromosome arms and centers. Our data indicated the complicated function and mechanisms of the

**Fig. 2 | Distinct chromodomain protein-histone modification association maps on chromosome arms and centers. a** Heatmap showing percentage of chromodomain protein peaks covered by histone modifications in chromosome arms. (**b**–**e**) Heatmaps comparing heterochromatin-enriched histone modifications (H3K9me2, H3K9me3, H3K27me3) and euchromatin-enriched histone modifications (H3K4me3, H3K36me3, H3K79me3) with the indicated chromodomain proteins in chromosome arms. Each horizontal line shows a target of the indicated chromodomain protein. *n*, number of targets. Raw reads were normalized to bins per million mapped reads (BPM). The color scale shows the range of heatmap intensities. **f** A summary of the interaction of chromodomain proteins with histone modifications on chromosome arms. "Graded histone modification occupied" for a pair of chromodomain protein and histone modification fell into 3 categories: grade 3 = "prominent" (red), grade 2 = "detectable" (green), and grade 1 = "weak" (purple). See Methods for details. **g** Distribution of chromodomain protein ChIP-seq peaks on different genomic features (promoters, exons, introns, etc.) across distinct broad chromosome domains. **h** Bar plot showing the association of chromodomain proteins with repetitive elements across distinct broad chromosome domains on chromosomes.

interaction of chromodomain proteins and histone modifications in vivo.

### Functional annotation of chromodomain proteins

To investigate the function of chromodomain proteins, we annotated the binding targets of each protein.

Most chromodomain proteins were enriched on protein-coding genes (Supplementary Fig. 14a). UAD-2 was highly enriched in piRNA genes. We quantified UAD-2 ChIP-seq signals on piRNA and protein-coding gene targets, respectively. The UAD-2 ChIP-seq signals on piRNA targets were significantly higher than those on protein-coding genes (Supplementary Fig. 14b). The result was consistent with our previous work showing that UAD-2 mediates heterochromatin-directed piRNA expression (Supplementary Fig. 14a, b)[24]. Then, we performed pathway enrichment analysis of the gene targets and divided the pathways into "Development", "Aging", "Stress response", "Cell cycle", and "Biosynthesis & Metabolism" related terms (Supplementary Fig. 14c). Most chromodomain proteins were enriched in a large number of biological pathways (Supplementary Fig. 14c), suggesting general roles of the chromodomain proteins in these biological processes. UAD-2 was not enriched in any specific GO (Gene Ontology) terms. CEC-8 and CEC-10 were also depleted from many specific GO (Gene Ontology) terms (Supplementary Fig. 14c).

A portion of each chromodomain protein's peaks overlapped with at least one repeat sequence (19.99-53.66%) (Fig. 2h). To test whether chromodomain proteins are biased toward specific repetitive sequence families, we classified 84,972 individual repetitive elements into 111 repeat families, which were further classified by sequence type (e.g., DNA transposon, retrotransposon, satellite, or unknown)[35]. Among the 111 repeat families, forty-two were mostly DNA transposons, including the transposase and Helitron families, and were bound by at least one chromodomain protein (Supplementary Figs. 15 and 16a–c).

### Epigenetic landscape of the chromodomain proteins

In eukaryotes, a plethora of histone modifications, writers, readers, erasers, and remodelers cooperate to translate the chromatin epigenetic landscape into transcriptional activation or repression and genome stability modulation[9].

To investigate the function of chromodomain proteins in the context of the epigenetic regulatory network, we downloaded previously published ChIP-seq datasets of a number of histone modification factors and epigenetic regulators. We first compared the correlation of the genome-wide distribution of these factors with that of the chromodomain proteins. We conducted hierarchical clustering of the genome-wide correlation coefficients and identified three major groups (Fig. 3a). The comparative analysis of all datasets was consistent with their putative functional relationships. Marks that are known to act in related pathways, such as transcriptional activation or repression, were highly correlated and clustered together. Group A contained repressive marks, including H3K9me3 and H3K27me3. Group B contained H3K9me2 and regulating factors such as MET-2, LIN-13, LIN-61, and HPL-2[14,17,35,37–39]. LEM-2 (ceMAN1), a nuclear membrane protein associated with heterochromatin, was also included[40]. In addition, H3K9me1 in embryos, H3K36me1 in adults, and H3K27me1 in embryos were included in this group. Remarkably, most of the chromodomain proteins studied in this work (10 out of 13) were highly correlated with each other and clustered into Group B, suggesting similar functions of these chromodomain proteins on repressive chromatin. Group C was mainly composed of active histone marks and related proteins. Of these, H3K4me1/2/3 were clustered in the same subcluster, whereas H3K36me2/3 and H3K79me2/3 shared more similar profiles. The chromodomain proteins CHD-3 and LET-418 were clustered with H3K4me1/2/3. MRG-1 and CEC-7, two members of the Msl-3 family, were contained in the subgroup, including H3K36me2/3 and H3K79me2/3. The finding that MRG-1 was highly correlated with H3K36me2/3 was consistent with previous research showing that MRG-1 and its homologs in yeast and humans are deposited in euchromatic regions bearing H3K36me[31,41].

To further identify specific functional combinations or communities of the chromodomain proteins, histone modifications and their regulators, we generated a colocalization network based on the overlap of binding sites observed in the ChIP-seq datasets (Fig. 3b). We calculated the overlap significance between the binding sites of each pair of factors and the percentage of overlapping sites. We then identified strong and moderate colocalization significance of the factors (see Methods). Overall, we identified 8 communities in the network (Fig. 3b, Supplementary Fig. 17a–c), which is similar to that of the hierarchical clustering approach. Most of the chromodomain proteins were highly interconnected with each other and exhibited a strong correlation with hallmarks of repressive chromatin, especially H3K9me2 and H3K27me3. CHD-3 and LET-418 displayed a high propensity for H3K4me2/3-related factors. MRG-1 and CEC-7 were present in the subnetwork containing H3K36me2/3 and H3K79me2/3. Detailed information on the epigenetic network is listed in Supplementary Data 3.

Taken together, we delineated an epigenetic regulatory landscape involving chromodomain proteins by combining clustering and network analysis of a plethora of ChIP-seq datasets. Our analysis sheds light on the potential diverse roles and mechanisms of chromodomain proteins in epigenetic regulatory networks. The data will facilitate the elucidation of the biological roles and regulation of many uncharacterized chromodomain proteins.

### Distinct subcellular localization of the chromodomain proteins

We then profiled the subcellular localization of each chromodomain protein in mitotic, meiotic cells, oocytes, early and late embryos (Supplementary Figs. 18a, 19, Supplementary Table 7) and assigned them to one or more of the 3 subcellular localization patterns: nucleoplasm, nucleolus, and nuclear puncta. Eight chromodomain proteins, CHD-1, CHD-3, CHD-7, LET-418, CEC-7, CEC-8, HERI-1, and CEC-10, were mainly localized to the nucleoplasm (Fig. 4a, Supplementary Figs. 18a, 19, Supplementary Table 7). Interestingly, four heterochromatin-related proteins, CEC-5, CEC-8, SET-31, and UAD-2, accumulated in certain subnuclear compartments (Fig. 4b–c, Supplementary Figs. 18a, b, 19, Supplementary Table 7)[24]. UAD-2 localized to chromatin and formed nuclear piRNA foci in the mitotic zone and early meiotic cells (Fig. 4b and Supplementary Fig. 18a)[24]. In embryos, UAD-2 colocalized with chromosomes during mitotic metaphase (Supplementary Fig. 19). Our

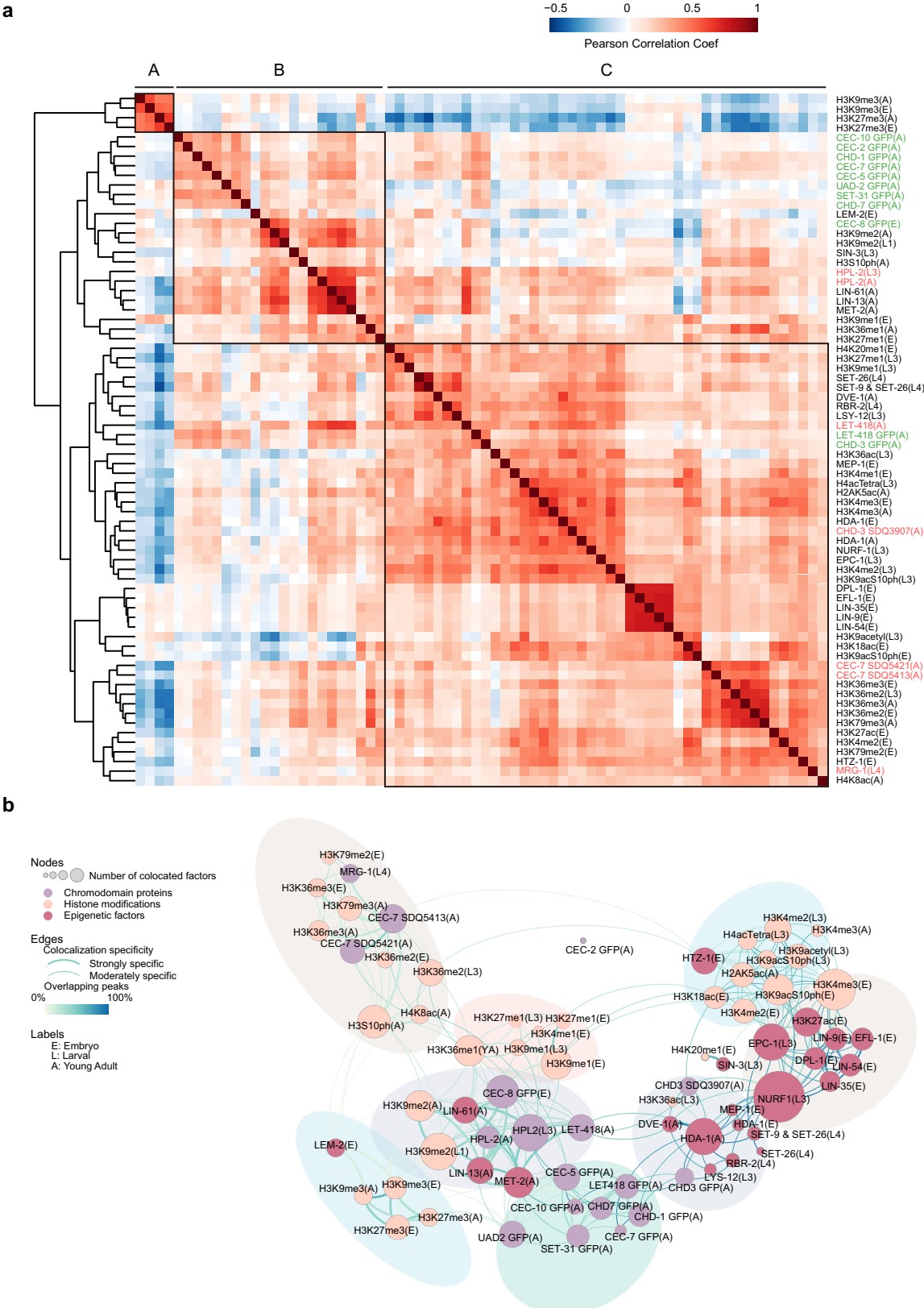

previous work showed that UAD-2, together with an upstream sequence transcription complex (USTC), colocalized with piRNA foci that were associated with piRNA clusters on chromosome IV[24,25].

The subcellular localization of CEC-5 diverged in different tissues or developmental stages. In mitotic zone or early meiotic cell, CEC-5 mainly localized to nuclear puncta on chromosomes (Fig. 4b,

Supplementary Fig. 18a). In oocytes, CEC-5 was mainly enriched in nucleolus (Supplementary Fig. 20a, b). In embryos, CEC-5 formed nuclear puncta on chromosomes (Supplementary Fig. 19). In late embryos, CEC-5 also colocalized with nucleolar marker FIB-1 (Fig. 4c, Supplementary Fig. 20b). In addition, CEC-5 constantly localized to nucleolus in somatic cells (Supplementary Fig. 20a, b). Consistently,

**Fig. 3 | An epigenetic regulatory network of chromodomain proteins.**
**a** Hierarchical clustering of chromodomain proteins and epigenetic regulators based on the pairwise Pearson correlations of signals of ChIP-seq peaks (see Methods for details). (Red) Positive correlations; (blue) negative correlations. The development stage of animals for each sample is noted as E (embryos), L (larval), and A (adult). Sample names are labeled by different colors. Chromodomain protein samples generated in this work (green), chromodomain protein samples downloaded (red), histone modifications, and epigenetic regulators (black). **b** A colocalization network across the genome of chromodomain proteins and epigenetic factors. In the network, nodes indicate individual chromodomain proteins (purple), histone modifications (light pink) and other epigenetic regulators (dark pink); subnetworks identified are indicated by shadows in different colors; edge colors depict the percentages of overlap between the nodes and weight the colocalization specificity between two factors (see Methods for details). The development stage of animals for each sample is noted as E (embryos), L (larval), and A (adult).

the ChIP-seq assay showed that CEC-5 occupied the 18S, 5.8S, and 26S rDNA genes (Supplementary Fig. 20c).

SET-31 was expressed in the germline and enriched on chromosomes. In mitotically proliferating cells, SET-31 colocalized with mCherry::H2B (Supplementary Fig. 18b) and formed nuclear puncta in germ cells (Fig. 4b, Supplementary Fig. 18a).

CEC-8 was also localized in the nucleolus. In a small proportion of embryo cells (~15%), we observed that CEC-8 formed two distinct nuclear foci that colocalized with the nucleolus marker mCherry::RRP-8 (Fig. 4c, Supplementary Fig. 20b). We did not detect significant binding of CEC-8 to either 18S, 5.8S, or 26S rDNA genes in the ChIP-seq experiment (Supplementary Fig. 20c).

The GFP-tagged transgenes could provide valuable reporter systems for investigating the function of chromodomain proteins, spatial distribution and condensation of heterochromatin and nucleolar regulation.

### A candidate-based RNAi screening to identify regulators of chromodomain proteins

Previously, we selected 239 genes involved in chromatin regulation and histone modification and designed a candidate-based RNAi screen to search for regulators of UAD-2 and piRNA transcription[24]. We successfully identified a number of genes, including *mes-2/3/4/6, isw-1* and *mrg-1*, etc., which are required for piRNA focus formation and piRNA transcription[24].

Here, to identify regulators of other chromodomain proteins, we used their subcellular localization as reporters to search for factors regulating the expression and localization of chromodomain proteins by fluorescence microscopy. Nine GFP-fused chromodomain proteins were investigated (Fig. 4d). Although we did not notice pronounced subcellular change of other chromodomain proteins by RNAi knocking down the epigenetic factors, we observed that knocking down *uba-2 and ubc-9* and the mutation of *met-2* significantly inhibited the nuclear puncta formation of CEC-5 (Fig. 4e–g).

UBA-2 is an E1 protein, and UBC-9 is a single E2 SUMO-conjugating enzyme in *C. elegans*[42]. Modification by the small ubiquitin-related modifier (SUMO), known as SUMOylation, includes distinct enzymatic pathways that conjugate SUMO to target proteins. Recent studies have linked SUMOylation to several chromatin regulation processes[43]. The observation that SUMOylation affects CEC-5 localization (Fig. 4f, g) suggested that SUMOylation may change the chromatin environment and modulate the function of CEC-5.

Strikingly, mutation of MET-2 disrupted the nuclear puncta formation of CEC-5 in the germline, whereas mutation of SET-25 did not affect CEC-5 localization (Fig. 4f, g). In *met-2* mutants, CEC-5 is significantly enriched in nucleoli. MET-2 is the homolog of mammalian SETDB1 that mediates mono- and dimethylation of H3K9. SET-25 deposits H3K9me3 marks in a MET-2-LIN-61- and NRDE-3-dependent manner[39]. These data suggested that CEC-5 may be regulated by H3K9me1/2.

Notably, there appeared to be different patterns of CEC-5 delocalization in *uba-2* and *ubc-9* RNAi animals compared to *met-2* mutants. One large and 1-2 small puncta still remained in *uba-2/ubc-9* RNAi, whereas CEC-5 puncta totally disappeared in *met-2* animals, suggesting that SUMOylation may have different effect on CEC-5

occupancy at certain genomic loci compared to MET-2 and H3K9me1/2.

To further assess the roles of MET-2 and H3K9me1/2 in the regulation of CEC-5, we performed ChIP-seq experiments of GFP::CEC-5, GFP::CEC-5;*met-2*, GFP::CEC-5;*set-25* and GFP::CEC-5;*met-2;set-25* in young adult animals. Notably, GFP::CEC-5 and GFP::CEC-5;*set-25* exhibited similar binding patterns; CEC-5 mainly bound chromosome arms and was associated with heterochromatin (Fig. 5a–c). However, in the *met-2* single mutant and *met-2;set-25* double mutant, CEC-5 significantly reduced its association with chromosome arms and heterochromatin, whereas the association of CEC-5 with chromosome centers and euchromatin was increased (Fig. 5a, d, e). We compared the ChIP enrichment of CEC-5 for wild-type and mutant animals (*met-2* single mutant, *set-25* single mutant and *met-2;set-25* double mutant) at all of the peaks normally called in wild-type animals. 779 out of the 3558 targets showed dramatically reduced CEC-5 binding in *met-2* single mutant and *met-2;set-25* double mutant, but not in *set-25* single mutant. In addition, the reduced CEC-5 binding targets were occupied by H3K9me2 (Fig. 5f). These data suggested that MET-2 and H3K9me1/2 were required for the proper association of CEC-5 with chromatin. Alternatively, the fact that in *met-2* single mutant and *met-2;set-25* double mutant, CEC-5 remained bound to the majority of CEC-5 peaks (Fig. 5b–f) found in wild-type animals may suggest that CEC-5 might also bind to chromatin independently of H3K9me status. Since the microscopy experiments (Fig. 4f) and genome tracks (Fig. 5a) suggested that *met-2* was required for CEC-5 binding, the ChIP signal observed in heatmaps (Fig. 5b–f) could be artifacts from the way of representing data, which requires further investigation.

In addition, we also identified 134 co-upregulated and 276 co-downregulated CEC-5 targets that were enriched in both *met-2* single mutant and *met-2;set-25* double mutant (Fig. 5g, h, j). Interestingly, the co-upregulated sites were uniformly distributed from chromosome centers to arms (Fig. 5i), whereas the co-downregulated targets were mainly distributed on chromosome arms (Fig. 5k). However, because of the lack of spike in control, it is unclear whether the appearance of these peaks is due to an increased binding of CEC-5 to these regions, or whether there is a general loss of chromatin association of CEC-5, and the signal observed in the ChIP-seq is noise amplified due to PCR and normalization.

Together, these results suggested that MET-2 and likely H3K9me1/2 were required for the proper association of CEC-5 with heterochromatin.

### The chromodomain of CEC-5 binds H3K9me0/1/2/3 in vitro
To assess whether CEC-5 directly binds to histones, we expressed and purified the GST-fused chromodomain-containing fragment of CEC-5 (CEC-5 CD, 51-172 aa). The fragment was incubated with biotin-labeled nonmethylated histone H3 peptides or methylated H3 peptides followed by precipitation with streptavidin agarose beads. The pelleted proteins were then resolved by SDS–PAGE and stained with Coomassie blue (Fig. 6a, b). As controls, GST and a GST-fused CEC-4 chromodomain-containing fragment (CEC-4 CD, 25-141 aa) were used[24,30]. The CEC-4 chromodomain has been shown to bind H3K9me1/2/3 in vitro[24,30]. Consistent with previous results, CEC-4 bound to H3K9me1/2/3 peptides (Fig. 6c, d) but not H3K9me0, H3K27me0, or

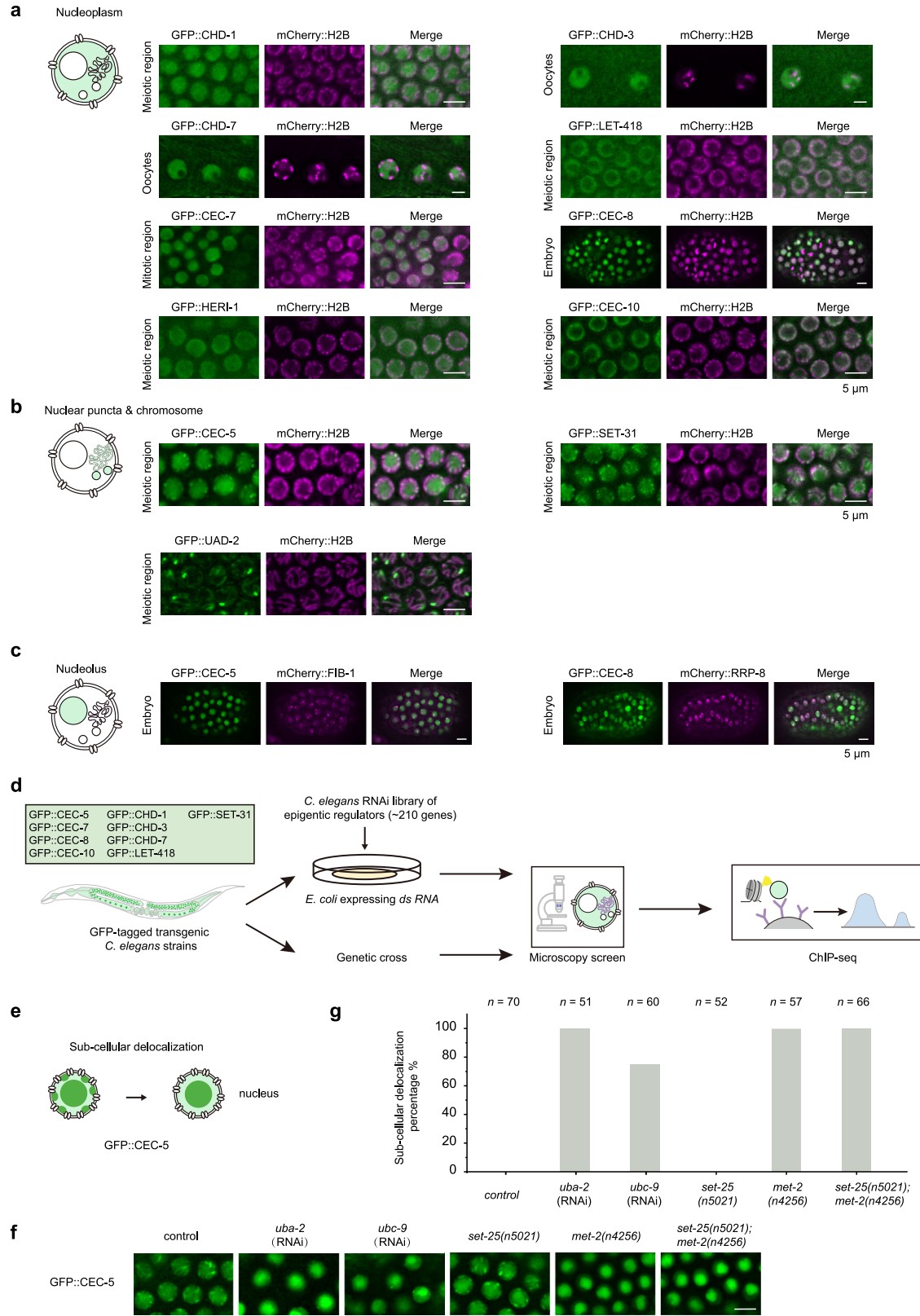

**Fig. 4 | A candidate-based RNAi screening to search regulators of chromodo-main proteins. a–c** The three distinct subcellular localization patterns of chro-modomain proteins. **a** Nucleoplasm in the germline. **b** Nuclear puncta and chromosomes in the germline. **c** Nucleolus in embryos. The localization of a representative image for each chromodomain protein is shown (scale bar, 5 μm). The chromosome is marked by mCherry::H2B, and the nucleolus is marked by mCherry::FIB-1 or mCherry::RRP-8. **d** Schematic procedure of the candidate-based RNAi screening. The subcellular localization of each chromodomain protein is used as a reporter. A library of 217 genes involved in chromatin modification and histone modification were selected to be knocked down or knocked out. **e** Schematic diagram of the screening for CEC-5 regulators. **f** Images of representative germline nuclei of the indicated adult animals. Knockdown of *uba-2* and *ubc-9* or mutation of *met-2* suppressed CEC-5 nuclear foci formation. **g** Quantification of the subcellular delocalization percentage in the indicated animals; *n*, the number of animals tested.

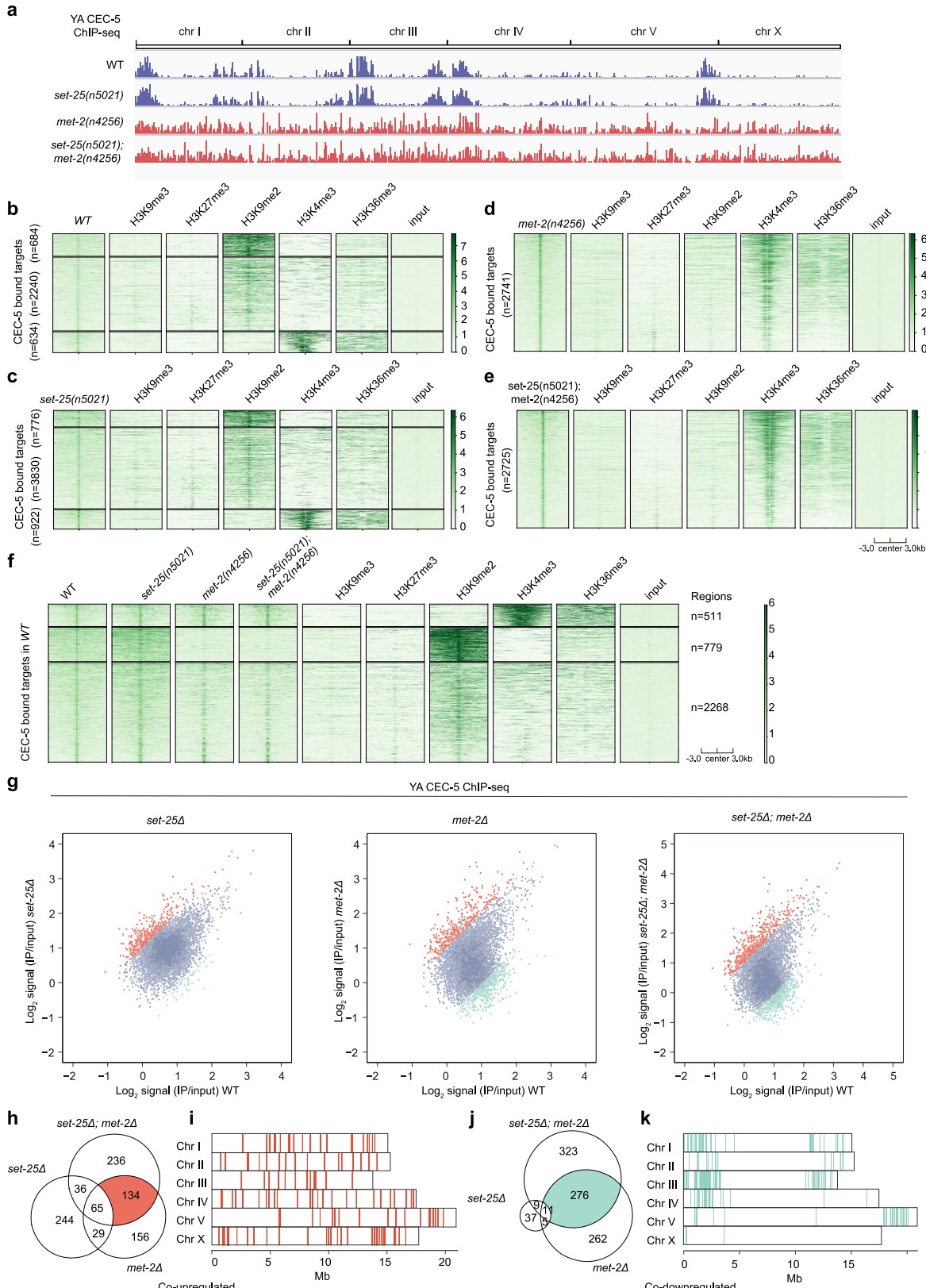

**Fig. 5 | The association of CEC-5 with chromatin depends on H3K9me1/2. a** ChIP-seq peak distribution of CEC-5 in the indicated adult animals. **b–e** Heatmaps comparing heterochromatin histone modifications (H3K9me2, H3K9me3, H3K27me3) and euchromatin histone modifications (H3K4me3, H3K36me3) with CEC-5 in the indicated animals. **f** Heatmap comparing CEC-5(*WT*), CEC-5(*set-25*), CEC-5(*met-2*) and CEC-5(*set-25;met-2*) ChIP-seq signals with heterochromatin histone modifications (H3K9me2, H3K9me3, H3K27me3) and euchromatin histone modifications (H3K4me3, H3K36me3, H3K79me3) at CEC-5(WT) targets. *n* = 3558. **g** Scatterplot comparing CEC-5 ChIP-seq signals in *set-25Δ*, *met-2Δ*, or *set-25Δ; met-2Δ* with wild-type animals. Differentially enriched peaks (>2-fold) for each genotype versus the wild type are highlighted in color (red, upregulated; blue, unchanged; green, downregulated). **h–k** Chromosome distribution of co-upregulated peaks (**h**, **i**, red) and co-downregulated peaks (**j**, **k**, green) in *met-2Δ*, and *set-25Δ; met-2Δ* compared to wild-type animals.

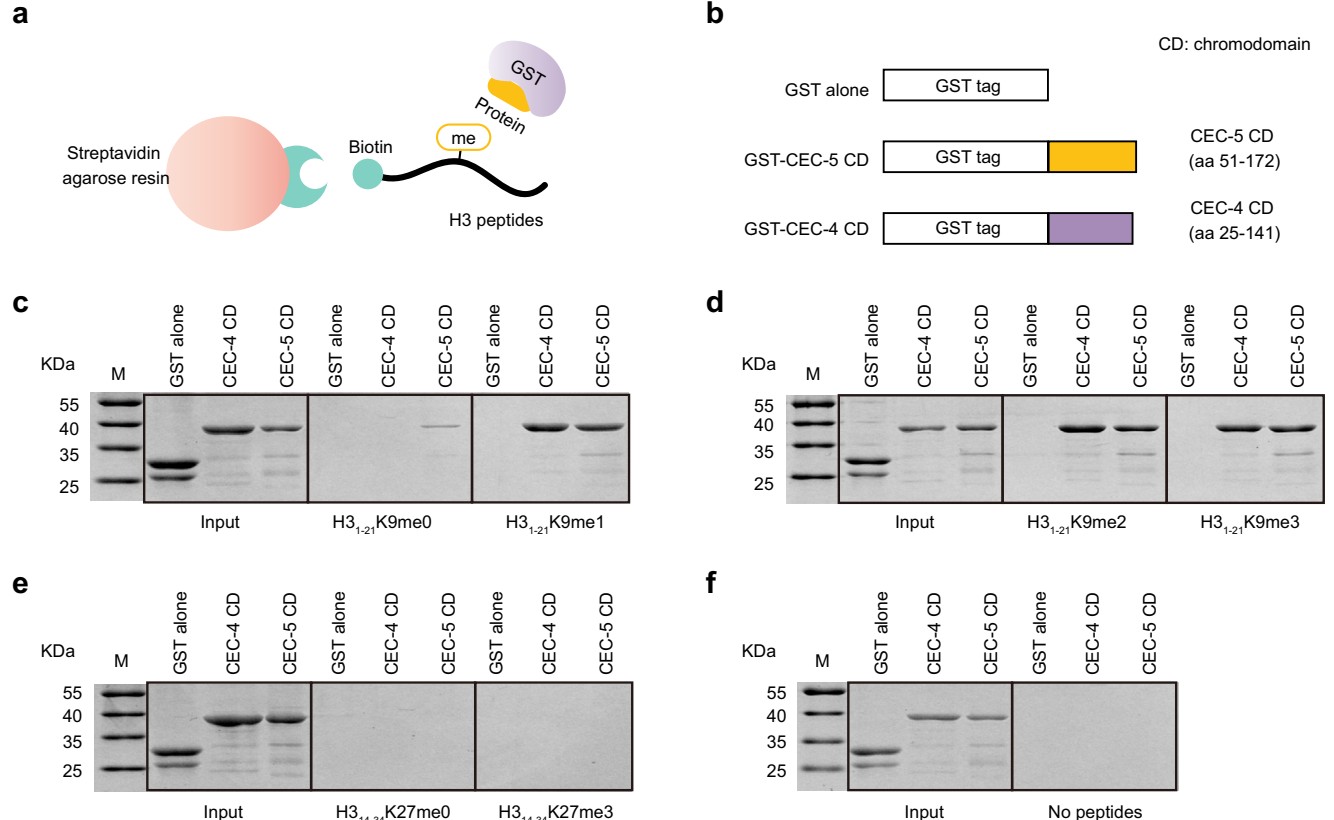

**Fig. 6 | The chromodomain of CEC-5 binds H3K9 peptides in vitro. a** Schematic diagram of the histone peptide pull-down assay. **b** Schematic diagram of the GST-tagged CEC-5 chromodomain (CEC-5 CD) and CEC-4 chromodomain (CEC-4 CD). **c**–**f** Coomassie blue-stained SDS–PAGE gels showing binding of the indicated proteins to biotinylated histone H3 peptides. GST alone served as a negative control. The CEC-4 chromodomain was used as a positive control. **c** H3 (1-21) K9me0 (middle) and H3 (1-21) K9me1 (right). **d** H3 (1-21) K9me2 (middle) and H3 (1-21) K9me3 (right). **e** H3 (14-34) K27me0 (middle) and H3 (14-34) K27me3 (right). **f** no peptides (right). Source data are provided as a Source Data file.

H3K27me3 peptides (Fig. 6e, f). CEC-5 did not bind either H3K27me0 or H3K27me3 peptide (Fig. 6e, f). Surprisingly, CEC-5 bound both unmethylated (me0) and methylated (me1/2/3) H3K9 peptides (Fig. 6c, d). Although the CEC-5 chromodomain exhibited low affinity to H3K9me0 in the in vitro binding assay, the in vivo dependency on *met-2* for binding suggested that CEC-5 protein binds methylated H3K9. In addition, the binding of CEC-5 to chromosomes was likely independent of SET-25 in vivo, by both ChIP-seq and subcellular localization. MET-2 is essential for ~80% of all the H3K9me3 catalyzed by SET-25 and in the absence of SET-25 these regions retain H3K9me2[38]. Our data suggested that CEC-5 recognizes H3K9me1/2 in vivo.

**Forward genetic screening identified that Arg124 in the chromodomain of CEC-5 was required for H3K9me1/2 binding**
To further understand the regulation of CEC-5, we performed a forward genetic screening via chemical mutagenesis to search for mutants in which the expression or localization of CEC-5 was altered by clonal screening (Fig. 7a, see details in methods). We isolated a single mutant from approximately one thousand haploid genomes, which disrupted the nuclear puncta formation of CEC-5 (Fig. 7b). We deep sequenced the mutant genome and identified a substitution of Arg124 with cysteine (*R124C*) in the chromodomain of CEC-5 (Fig. 7c). Notably, Arg124 was conserved in human HP1 (HP1β-CBX1, HP1γ-CBX3, HP1α-CBX5) and PC (PC3-CBX8) proteins, as well as in *C. elegans* CEC-4, CEC-5, and CEC-8, suggesting a critical role of the Arg124 residue (Fig. 7d). In addition, the *R124C* mutation reshaped the distribution of CEC-5 on heterochromatin and euchromatin (Fig. 7e). 746 out of the 3558 targets showed dramatically reduced CEC-5 binding in *R124C*. The

reduced CEC-5 binding targets were also occupied by H3K9me2 (Fig. 7f). To compare CEC-5 occupancy in *R124C* mutation and *met-2* mutation, we overlapped the reduced CEC-5 targets in *met-2* and *R124C* mutants. Strikingly, 745 out of the 746 reduced R124C targets were overlapped with those in *met-2* mutant (Fig. 7g). Taken together, these results suggested that the Arg124 residue in chromodomain played important roles for CEC-5 to recognize H3K9me1/2 in vivo.

Epigenetic alteration is one of the hallmarks of aging in many organisms[44]. Aged animals usually show dysregulated repressive heterochromatin[45–51]. Recent work has reported that the depletion of H3K9me1/2 by *met-2* mutation shortened the lifespan of *C. elegans*[52]. To test whether CEC-5 functions in aging and longevity, we assayed the lifespan of *cec-5* mutants. Both *cec-5(ust240)* and *cec-5(ust283)* were short-lived compared to wild-type N2 animals. Interestingly, *cec-5, met-2* and *daf-16* all exhibited similar shortened lifespans (Fig. 7h, i, Supplementary Fig. 21a–d).

To explore the link between CEC-5 and nucleolus, we quantified rRNA expression levels in *WT, met-2* and *cec-5* mutants by quantitative real-time PCR. However, we failed to detect pronounced alterations on mature rRNA (18 S rRNA, 5.8 S rRNA, and 26S rRNA) levels (Supplementary Fig. 20d). Moderate increase of pre-rRNA levels was detected in *met-2* and *cec-5* mutants (Supplementary Fig. 20e). The nucleolus morphology in either *met-2* or *cec-5* mutants were not noticeably changed either (Supplementary Fig. 20f).

Taken together, our data suggested that CEC-5, MET-2, and H3K9me1/2 were likely required for the normal lifespan of *C. elegans*. The interaction of CEC-5 occupancy on H3K9me-enriched genomic regions and nucleolus may be critical for CEC-5 function.

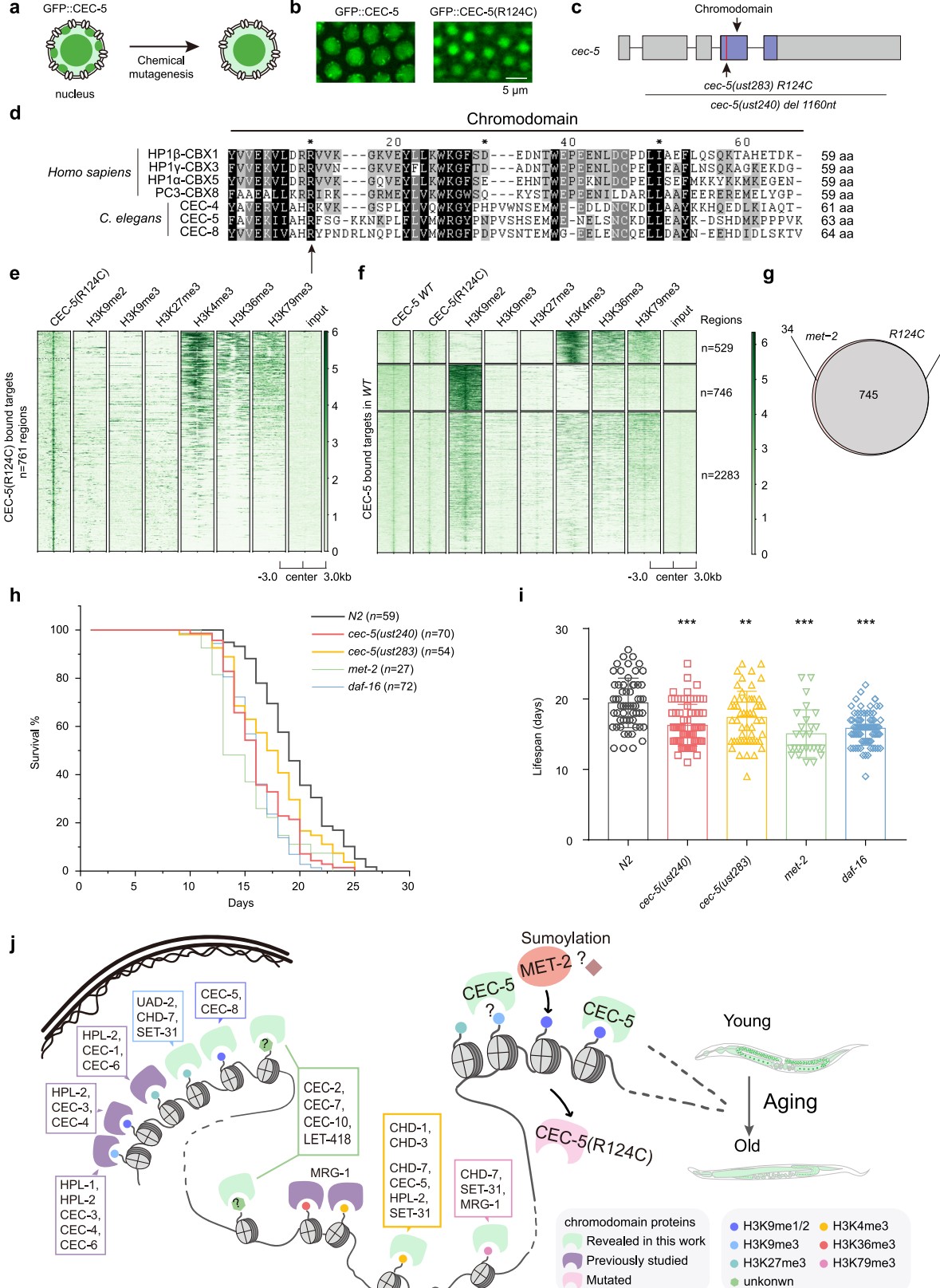

## Discussion

In this work, we generated a resource of chromodomain proteins in *C. elegans* by fluorescently tagging the chromodomain proteins using the CRISPR/Cas9 system and performing ChIP-seq, imaging, genetic screening, and functional examination. We presented a functional map of chromodomain proteins (Fig. 7j). We searched for epigenetic factors regulating chromodomain proteins and found that H3K9me1/2 was essential for CEC-5 subcellular localization and heterochromatin association. We conducted a forward genetic screening and identified a conserved residue in the CEC-5 chromodomain required for recognition of H3K9me1/2. *cec-5* is required for the normal life span of *C. elegans*.

**Fig. 7 | CEC-5 is required for normal lifespan by recognizing H3K9me1/2.**
**a** Schematic diagram of EMS screening for CEC-5 subcellular delocalization.
**b** Images of representative germline nuclei of the indicated adult animals.
**c** Schematic diagram of the *cec-5* exon (gray) and chromodomain (purple).
**d** ClustalW revealed the multiple sequence alignment of *C. elegans* CEC-4, CEC-5, and CEC-8 chromodomains with different *H. sapiens* HP1 and PC proteins.
**e** Heatmap comparing CEC-5(R124C) ChIP-seq signal with heterochromatin histone modifications (H3K9me2, H3K9me3, H3K27me3) and euchromatin histone modifications (H3K4me3, H3K36me3, H3K79me3) at CEC-5(R124C) targets. **f** Heatmap comparing CEC-5(WT) and CEC-5(R124C) ChIP-seq signals with heterochromatin histone modifications (H3K9me2, H3K9me3, H3K27me3) and euchromatin histone modifications (H3K4me3, H3K36me3, H3K79me3) at CEC-5(WT) targets. $n = 3558$.
**g** Venn diagram comparing reduced CEC-5 occupying targets in CEC-5(R124C) and *met-2*. (**h**) Survival curves of the indicated animals. *n*, the number of animals tested. See two replicates of lifespan experiments in Supplementary Fig. 20a–d. *n*, the number of animals tested. **i** Histogram displaying lifespan of the indicated animals. Mean ± s.e.m. of animals. The number of animals tested of each strain is indicated in Fig. 7h. Asterisks indicate significant differences using one-sided *t* tests. **0.001 < $p$ < 0.01; ***$p$ < 0.001. *cec-5(ust240)*, $p = 6.60 \times 10^{-8}$; *cec-5(ust283)*, $p = 0.00148$; *met-2*, $p = 2.76 \times 10^{-7}$; *daf-16*, $p = 8.00 \times 10^{-11}$. **j** A working model summarizing the association of chromodomain proteins with histone modifications in the *C. elegans* genome. Source data are provided as a Source Data file.

## Conservation of chromodomains in eukaryotes

The *C. elegans* genome encodes 21 proteins that contain chromodomains, 9 of which have been reported to have homologs in humans (Supplementary Table 1). Although the remaining chromodomain proteins in *C. elegans* have no defined homologs in humans, the chromodomain sequences of these proteins still show high similarity to those of human proteins (Supplementary Table 2)[3]. Interestingly, most of these worm-specific chromodomain proteins have a single chromodomain, without any accompanying catalytic domains. The complexity of worm-specific chromodomain proteins suggests that sequences flanking chromodomain or cofactors may modulate the substrate specificity or functions of the proteins. We identified a point mutation in the CEC-5 chromodomain critical for its enrichment in H3K9me1/2-associated heterochromatin. This residue is highly conserved in human HP1 (HP1β-CBX1, HP1γ-CBX3, HP1α-CBX5) and PC (PC3-CBX8) proteins, suggesting a conserved mechanism of H3K9me recognition during evolution.

## Systematically fluorescent tagging of chromodomain proteins

In the last decade, CRISPR/Cas9 technology has revolutionized gene editing to facilitate efficient and precise fluorescent tagging in many organisms. The engineered cells enable examination of protein expression and function in near-native environments. In addition, the transparent and short-life-cycled *C. elegans* provides a system to measure the subcellular localization and regulation of proteins under physiological conditions on a life-long scale. The systematic fluorescent tagging of chromodomain proteins provides a resource to investigate the mechanism and function of epigenetic modifications. For example, CEC-5 is localized to the nucleolus and associated with rDNA, suggesting an unknown role of epigenetic modification of histones in rRNA regulation in *C. elegans*[53–55].

Using the GFP::CEC-5 strain as a reporter, we found that knocking down the SUMOylation E1 protein *uba-2* or the E2 SUMO-conjugating enzyme *ubc-9* disrupted the nuclear puncta of CEC-5. SUMOylation has been shown to promote the formation and maintenance of silent heterochromatin in yeast and animals[43,56–58]. Our results suggest an underlying mechanism of SUMOylation-controlled heterochromatin regulation.

## The epigenetic regulatory network of chromodomain proteins

In eukaryotic cells, chromosomes are segregated into domains of heterochromatin and euchromatin with distinct functional properties. Heterochromatin is highly condensed, gene poor, rich in transposons or other parasitic genomic elements and generally transcriptionally silent, whereas euchromatin is less condensed, gene-rich, and more easily transcribed[59–61]. These two silent and active compartments of chromatin differ in characteristic posttranslational modifications (PTMs) of the core histones, as well as in the incorporation of specific histone variants, linker histones, and nonhistone proteins[29]. Heterochromatin is enriched for H3K27me3 and three methylated states of H3K9, whereas euchromatic domains are marked by H4 acetylation, H3K4me1/2/3, H3K36me1/2/3, and H3K79me1/2/3[12,62]. Chromodomain proteins bind both heterochromatin and euchromatin-enriched methyl-lysine of histones. The single chromodomain usually recognizes H3K9me2/3 or H3K27me2/3. Paired tandem chromodomains in CHD family proteins are able to bind H3K4me1/2/3. In addition, the chromo barrel domain, a chromo-like domain, can interact with H3K36me2/3.

In this work, we delineated putative histone modifications recognized by chromodomain proteins and related epigenetic factors via systematic analysis of ChIP-seq datasets. Furthermore, by combining genetic screening and in vitro biochemical experiments, we found that CEC-5 binds to H3K9me1/2. Nevertheless, further investigations are required to reveal the direct (or physical) interactions of chromodomain proteins with histones and epigenetic factors. Therefore, a comprehensive interactome (by IP-MS or in vitro pull-down assay) of chromodomain proteins will facilitate the understanding of the mechanism and function of chromodomain proteins. GFP-tagged chromodomain proteins could be a promising tool to search for factors modulating heterochromatin.

## Complicate functions of chromodomain proteins

The major function of chromodomain proteins lies in gene expression regulation, genome stability, and three-dimensional genome architecture. In addition to regulating protein-coding genes, our work suggested that chromodomain proteins also participate in the control of noncoding sequences. For example, UAD-2 is required for the recruitment of the upstream sequence transcription complex (USTC) to piRNA clusters and promotes piRNA production[24]. HPL-2 and LET-418 are enriched at repetitive elements and prevent aberrant expression of the sequences[35]. Furthermore, nearly all of the tested chromodomain proteins bind to repetitive sequences, implying the general function of chromodomain proteins in genome stability surveillance.

In metazoans, heterochromatic and euchromatic compartments can be distinguished by their spatial organization and association with the nuclear membrane-associated lamina. Recent studies have identified that the perinuclear anchoring of lamina-associated domains in *C. elegans* is facilitated by three chromodomain proteins, CEC-4, MRG-1, and HPL-2[3,30,31]. Whether other chromodomain proteins participate in the process remains to be determined.

## Chromodomain proteins regulate lifespan and aging

Aging is a complex multifactorial biological process in all living organisms. Among the characterized hallmarks of aging, epigenetic alterations represent a crucial mechanism[63–65].

The association between aging and dysregulated repressive heterochromatin has been observed across species, from yeast to humans. In yeast, loss of transcription silencing contributes to aging-related sterility[44]. In flies and worms, heterochromatin levels positively correlate with lifespan, and aging is associated with the deterioration of heterochromatin[45–48]. In mammals, the loss of heterochromatin markers, such as H3K9me, is associated with aging and premature aging diseases[49–51]. Nevertheless, the aging-associated gain of repressive histone modifications has also been observed. For example, in flies, H3K9me3 levels were found to increase in the aging brain, although they decreased in the aging intestine[66,67]. In mice, the levels of H3K27me3 are reduced in senescent fibroblast cells but elevated in the

brain of a mouse model of accelerated aging[68,69]. In addition, the active transcription mark H3K4me has been shown to be essential for lifespan suppression[70,71].

Our data showed that the loss of CEC-5, an H3K9me1/2 reader, and MET-2, the H3K9me1/2 writer, shortens the lifespan of *C. elegans*[52]. Notably, the *R124C* point mutation on the CEC-5 chromodomain, which disrupted CEC-5 binding to chromatin, also shortened the lifespan of *C. elegans*. How H3K9me and CEC-5 modulate aging and lifespan requires further investigation.

### Comparison of ChIP-seq datasets

In this work, we compared chromodomain protein ChIP-seq datasets obtained using GFP antibody (ab290) in our experiments and those previously published CHD-3, LET-418, and CEC-7 datasets. CHD-3 GFP(A) and published CHD-3 SDQ3907(A) both showed that CHD-3 associated with H3K4me3 (Figs. 2f and 3a, b, Supplementary Fig. 9b).

Both LET-418 GFP(A) and LET-418(A) ChIP-seq datasets showed that LET-418 was related to H3K4me3 (Figs. 2f and 3a, b, Supplementary Fig. 9b). However, LET-418(A) but not LET-418 GFP(A) showed that LET-418 was also correlated with H3K9me2 (Figs. 2f and 3b). Previous work (the LET-418(A) data) revealed that LET-418, HPL-2, LIN-13, LIN-61, and MET-2 were all associated with H3K9me2 and repetitive elements whereas LET-418 exhibited generally lower enrichment on repetitive sequences than the other factors[35]. We speculated that LET-418 may recognize both H3K4me3 and H3K9me2. The discrepancy might be caused by different antibodies used (ab290 vs. Q3861) or worm culture conditions (OP50 vs. HB101 *E. coli*).

CEC-7 GFP(A) suggested that CEC-7 may associate with H3K4me3 (Fig. 2f, S9b). Although CEC-7 SDQ5421(A) revealed weak association with H3K4me3 at chromosome centers, CEC-7 SDQ5413(A) and CEC-7 SDQ5421(A) showed that CEC-7 may mainly associate with H3K36me3 and H3K79me3 (Figs. 2f and 3a, b, Supplementary Fig. 9b). The CEC-7 SDQ5413(A) and CEC-7 SDQ5421(A) datasets were downloaded from modENCODE project, using antibody SDQ5413 and SDQ5421, respectively. The reason of discrepancy is unclear.

## Methods

### Strains

Bristol strain N2 was used as the standard wild-type strain. All strains were grown at 20 °C unless otherwise specified. The strains used in this study are listed in Supplementary Table 8.

### Construction of transgenic strains and mutants

For chromodomain protein transgenes, endogenous promoter sequences, UTRs, and ORFs of chromodomain genes were PCR-amplified with the primers listed in Supplementary Table 9. The coding sequence of *3xflag::gfp* and a linker sequence (GGAGGTG-GAGGTGGAGCT) (inserted between the ORFs and *3xflag::gfp*) were PCR-amplified with the primers 5′-atggactacaaagaccatgacgg-3′ and 5′-AGCTCCACCTCCACCTCCTTTG-3′. The vector was PCR-amplified from PCFJ151 using the primers 5′-tgtgaaattgttatccgctgg-3′ and 5′-caCACGTGctggcgttacc-3′. Fragments and the vector were fused using a ClonExpress MultiS One Step Cloning Kit (Vazyme C113-02, Nanjing). Chromodomain protein transgenes were integrated into the *C. elegans* genome locus of each gene in situ by using the CRISPR/Cas9 system[72–74]. The injection mix contained PDD162 (50 ng/μl), plasmid containing chromodomain protein transgenes (50 ng/μl), pCFJ90 (5 ng/μl) and two sgRNAs (30 ng/μl). The mix was injected into young adult N2 animals. Gene deletions were also conducted using the CRISPR/Cas9 system. The sg-RNA sequences are listed in Supplementary Table 10.

### Generation of the phylogenetic tree

Protein sequences were obtained from Wormbase.org and UniProt.org. There are 26 sequences in *C. elegans* (different isoforms were contained), and 36 sequences in human (only one isoform of the same

protein was chosen). Protein sequences were aligned using ClustalW using MEGA-X[75]. The phylogenetic tree was inferred by using the Neighbor-Joining method. The Poisson model was chosen as the substitution model and the Uniform Rates were used to model evolutionary rate differences among sites. Bootstrap analysis was based on 1000 replicates using the Neighbor-Joining method.

### Microscopy and imaging

Images were collected on a Leica DM4 B microscope by using Leica LAS X Microscope Software. Gonads were dissected in 0.4 × M9 buffer supplemented with 0.1 M sodium azide. Quantification of GFP intensities was performed by ImageJ (v1.8.0). The relative GFP intensity (relative to the average level of GFP::CEC-5 at mitotic region) of mitotic, transition zone, pachytene, diplotene, and oocyte zones was calculated, respectively. Four independent replicates were used.

### Chromatin immunoprecipitation (ChIP)

ChIP experiments were performed as previously described[76]. Worm samples in the adult stage were crosslinked in 2% formaldehyde for 30 min. Fixation was quenched with 0.125 M glycine for 5 min at room temperature. Samples were sonicated for 20 cycles (30 s on and 30 s off per cycle) at medium output with a Bioruptor 200. The lysates were precleared and immunoprecipitated with 1.5 μL (1.5:1000) of a rabbit anti-GFP antibody (Abcam, ab290) overnight at 4 °C. Chromatin/antibody complexes were recovered with DynabeadsTM Protein A (Invitrogen, 10002D) followed by extensive sequential washes with 150 mM, 500 mM, and 1 M NaCl. Crosslinks were reversed overnight at 65 °C. The input DNA was treated with RNase (Roche) for 30 minutes at 65 °C, and all DNA samples were purified using a QIAquick PCR purification kit (Qiagen, 28104).

### ChIP-seq

The DNA samples from the ChIP experiments were deep sequenced at Novogene Bioinformatics Technology Co., Ltd. (Tianjin, China). Briefly, 10–300 ng of ChIP DNA was combined with End Repair Mix (Novogene Bioinformatics Technology Co., Ltd. (Tianjin, China)) and incubated for 30 min at 20 °C, followed by purification with a QIAquick PCR purification kit (Qiagen). The DNA was then incubated with A-tailing mix for 30 min at 37 °C. The 3′-end-adenylated DNA was incubated with the adapter in the ligation mix for 15 min at 20 °C. The adapter-ligated DNA was amplified through several rounds of PCR amplification and purified in a 2% agarose gel to recover the target fragments. The average length was analyzed on an Agilent 2100 Bioanalyzer instrument (Agilent DNA 1000 Reagents) and quantified by qPCR (TaqMan probe). The libraries were further amplified on a cBot system to generate clusters on the flow cell and sequenced via a single-end 50 method on a HiSeq1500 system.

### ChIP-seq data downloaded

ChIP-seq datasets of histone modifications, epigenetic factors, and reported chromodomain proteins in *C. elegans* were downloaded from the NCBI GEO or ENCODE databases. The datasets used in this study are listed in Supplementary Data 1.

### ChIP-seq data analysis

ChIP-seq reads were aligned to the WBcel235 assembly of the *C. elegans* genome using Bowtie2 version 2.3.5.1[77] by Ben Langmead with the default settings. The SAMtools version 0.1.19[78] "view" utility was used to convert the alignments to BAM format, and the "sort" utility was used to sort the alignment files. ChIP-seq peaks were called using MACS2 version 2.1.1[79] with subcommand callpeak with defined parameters (-g ce -B -f BAM -q 0.01 -m 4 50). Size of most narrow peaks of chromodomain proteins was lower than 500 bp (more than 60%) besides MRG-1(L4) (~33.48%). For MRG-1(L4), ~36.50% peaks were larger than 1 kb. An overview of peak numbers and peak sizes was

provided in Supplementary Data 2. Representative genome tracks showing examples of peaks were included (Supplementary Fig. 6a–d). Besides, ChIP-seq broad peaks of chromodomain proteins were also called using MACS2 version 2.1.1 with subcommand callpeak with parameters (-g ce -B -f BAM –broad). We identified more broad peaks than narrow peaks for each chromodomain protein. The percentage of peaks >1 kb was nearly equal to peaks <500 bp (~30%). More than 90% narrow peaks were overlapped with broad peaks for each chromodomain proteins. In this work, we plotted heatmaps showing ChIP-seq signals near the center of each peak and analyzed the correlation of pairs of samples (Fig. 2b–e, S8d–g, S9a, 3a, 5g and Supplementary Figs. 10, S11). Calling peaks using MACS2 with the default narrow peak function has an advantage in the accuracy of signal intensity quantification across the peaks. Therefore, we used narrow peak calling for downstream analysis.

Deeptools subcommand bamCoverage (version 3.5.0) was used to produce bigWig track for data visualization with defined parameters (--binSize 20 --normalizeUsing BPM --smoothLength 60 --extendReads 150 --centerReads -p 6 2) from bam files. The Integrative Genomics Viewer genome browser[80] was applied to visualize signals and peaks. The heatmap was plotted with Deeptools subcommand plotHeatmap (version 3.5.0). The ChIP-seq peaks were annotated with the R package ChIPseeker (Yu et al., 2015).

### Quality control of ChIP-seq datasets

The quality of each chromodomain protein ChIP-seq dataset used in this work was assessed. Firstly, the ratio of reads falling within peak regions of each sample was calculated and defined as Fraction of Reads in Peaks (FRiP)[81]. Secondly, cross-correlation analysis was performed[81,82]. Two cross-correlation peaks are usually observed in a ChIP experiment, one corresponding to the read length ("phantom" peak) and one to the average fragment length of the library (ChIP peak). The normalized ratio between the fragment-length cross-correlation peak and the background cross-correlation (normalized strand coefficient, NSC) and the ratio between the fragment-length peak and the read-length peak (relative strand correlation, RSC), are strong metrics for assessing signal-to-noise ratios in a ChIP-seq experiment. High-quality ChIP-seq datasets tend to have a larger fragment-length peak compared with the read-length peak, whereas failed ones and inputs have little or no such peaks. Thirdly, MA plots were included to show the enrichment of each chromodomain protein versus input.

To compare chromodomain protein ChIP-seq datasets obtained using GFP antibody (ab290) with those published previously, correlation and IDR analysis for CHD-3 (CHD-3 GFP(A) and CHD-3 SDQ3907(A)), LET-418 (LET-418 GFP(A) and LET-418(A)) and CEC-7 (CEC-7 GFP, CEC-7 SDQ5413, and CEC-7 SDQ5421) were performed, respectively[81]. For correlation analysis, peaks called from any of the pairs of samples were used. The 95th percentile values were extracted from bigWig files using the Python package pyBigWig over each peak. The values were normalized to input signals, logarithm-transformed and then standardized to Z scores. Correlation coefficients were calculated by the cor [Pearson] function in R using the treated values. A two-sided $t$ test was performed. IDR (irreproducible discovery rate) analysis[81] was performed with a 0.05 threshold. Scatter plots of signal scores and ranks of peaks that overlap in each pair of samples were plotted. In addition, plots showing the estimated IDR as a function of different rank thresholds were also included.

### Analysis of association between chromodomain proteins and histone modifications

The genomic regions were defined as chromosome arms and centers based on previous works[12,83,84]. Briefly, recombination rates and H3K9 methylation enrichment were used to estimate the physical boundaries between the arms and central regions of each chromosome. Chromosome arms were characterized by high levels of recombination

rate and H3K9 methylation. Coordinates of arms and center regions of each chromosome were shown in Supplementary Table 4.

Heterochromatin was defined by occupancy of H3K9me2, H3K9me3, and H3K27me3, and euchromatin was determined by covering of H3K4me3, H3K36me3, and H3K79me3[12,85–88]. Association between chromodomain proteins and histone modifications was detected by correlation, heatmap, and peak overlapping analysis.

For correlation analysis, combined peaks were determined by peaks called from any of the chromodomain protein or histone modification ChIP-seq datasets. The overlapping peaks were merged ($n = 42,496$). Then the merged peaks were divided into chromosome arm ($n = 24,284$) and center peaks ($n = 18,212$). The 95th percentile values were extracted from bigWig files using the Python package pyBigWig over each chromosome arm and center peak. The values were normalized to input signals, logarithm-transformed, and then standardized to Z scores. Correlation coefficients were calculated by the cor [Pearson] function in R using the treated values. A two-sided $t$ test was performed.

For heatmap analysis in Fig. 2b–e, Supplementary Figs. 8d–g, 9a, the called peaks of each chromodomain proteins were used. Deeptools subcommand computeMatrix (version 3.4.3) was used to calculate score matrix for heatmap with defined parameters (reference-point --referencePoint center -b 3000 -a 3000 --skipZeros). The heatmap was plotted with Deeptools subcommand plotHeatmap (version 3.5.0). To identify heterogeneity of chromatin states of chromodomain protein targets, heatmap of each chromodomain protein was clustered. Parameter --kmeans was used to define the number of clusters. For each chromodomain protein, --kmeans 1, 2, and 3 were all applied. Plots displaying no heterogeneity of chromatin states within each cluster were selected. A plot with minimum kmeans number for each chromodomain protein from the selected plots was shown.

Significant overlapped peaks of chromodomain proteins and histone modifications were defined by using IntervalStats software package[89]. The method compared each single peak region from a 'query' experiment to the set of peak regions in a 'reference' experiment. P-values represents the significance of query peak proximity to the reference peak. Pairs of peaks were considered significantly overlapped with $P < 0.05$. Both chromodomain proteins and histone modifications were used as queries and references (Fig. 2a, Supplementary Fig. 8a–c). The percentage of overlap was listed in Supplementary Table 5. We used the percentage of PTM-covered peaks of each chromodomain protein for "graded histone occupied" evaluation. For many chromodomain proteins, PTM covered more than 30% of their peaks. However, for PTMs, chromodomain proteins rarely covered more than 30% (Supplementary Fig. 8b, c, Supplementary Table 5). We speculated that for a given PTM, there are a number of additional readers besides chromodomain proteins, such as Tudor and MBT, etc., may function redundantly to recognize the PTM.

"Graded histone modification occupied" for a pair of chromodomain protein and histone modification fell into 3 categories: grade 3 = "prominent", grade 2 = "detectable", and grade 1 = "weak". Grade 3 was defined as pairs meeting all of the 3 criteria: 1. "Pearson correlation coefficient ($r$) ≥ 0.300". 2. " ≥30% of significant peaks occupied by a certain histone modification (using IntervalStats software package with $P < 0.05$)". 3. "Overlap detected by heatmap analysis". Grade 2 meets 2 out of the 3 criteria and grade 1 meets any of the 3. For example, on chromosome arms, Pearson correlation coefficient of CEC-5 GFP(A) and H3K9me2, $r = 0.333$ (Supplementary Fig. 10); 44.3% peaks of CEC-5 were significantly occupied by H3K9me2 (Fig. 2a and Supplementary Table 5); Heatmap analysis found 1585 out of 1854 targets of CEC-5 were covered by H3K9me2 (Fig. 2b). Therefore, the association of CEC-5 and H3K9me2 on chromosome arms was classified as "grade 3 = prominent" (Fig. 2f and Supplementary Table 6).

### Definition of actively transcribed genes and silent genes

To identify actively transcribed genes and silent genes, we analyzed 2 repeats of mRNA-seq datasets of adult N2 worms. The average of Fragments Per Kilobase of exon model per Million mapped fragments (FPKM) of each gene was calculated. Actively transcribed genes were defined by "FPKM ≥ 1" and being occupied by any of the euchromatin marks (H3K4me3, H3K36me3, or H3K79me3). Silent genes were defined by "FPKM < 1" and being occupied by any of the heterochromatin marks (H3K9me2, H3K9me3, or H3K27me3).

### Cluster and heatmap analyses

The pipeline of cluster and heatmap analysis was adapted from a previously described method[90]. Groups of ChIP factors with similar genome-wide signals were determined using the hclust function in R (The R Development Core Team 2009), with pairwise correlation coefficients as the similarity measure. Combined peaks were determined by peaks called from any of the ChIP-seq datasets. The overlapping peaks were merged. The 95th percentile values were extracted from bigWig files using the Python package pyBigWig over each genomic locus of combined peaks. The values were normalized to input signals, logarithm-transformed and then standardized to Z scores. Correlation coefficients were calculated by the cor [Pearson] function in R using the treated values. The agglomeration method "complete" was used.

### Generation of an epigenetic regulatory network

To identify the specific combinations of chromodomain proteins with other epigenetic factors, we generated a colocalization network based on the overlap of binding sites observed in our ChIP-seq datasets using a previously described method[90]. In brief, we used the IntervalStats tool to compute significant overlaps between binding sites of each pair of factors and calculated the percentage of overlapping sites between the factors. Based on the percentage, we identified strong and moderate colocalization specificity of the factors. The Gephi tool was used to create and visualize the network (Fig. 3b)[91]. In the network, the weight of the edges represents how specifically two TFs are associated with each other, whereas the color indicates their percentage of overlapping binding sites. From the network, we identified clusters of strongly correlated chromodomain proteins and epigenetic factors. To highlight these clusters, we further partitioned the network into 8 subnetworks in different colors (Fig. 3b) using an algorithm developed by Blondel et al. and implemented in Gephi[90,91]. Detailed information on the epigenetic regulatory network is listed in Supplementary Data 3.

### Candidate-based RNAi screening

RNAi experiments were performed at 20 °C by placing synchronized embryos on feeding plates as previously described[92]. HT115 bacteria expressing the empty vector L4440 (a gift from A. Fire) were used as controls. Bacterial clones expressing dsRNAs were obtained from the Ahringer RNAi library and sequenced to verify their identity. All feeding RNAi experiments were performed for two generations except for sterile worms, which were RNAi treated for one generation. The genes in the epigenetic RNAi library used in this study are listed in Supplementary Data 4. Images were collected using a Leica DM4 B microscope.

### Recombinant protein expression and purification

The CEC-4 chromodomain domain (amino acids 25–141) and CEC-5 chromodomain (amino acids 51–172) were PCR amplified, cloned into a plasmid (pET-28a-N8×H-MBP-3C vector), and expressed in Escherichia coli Rosetta cells (Novagen). The recombinant proteins were affinity purified through GST tag binding to amylose resin (BioLabs) according to the manufacturer's instructions.

### Histone peptide pull-down assay

The histone peptide pull-down assay was performed basically as described previously[24]. Briefly, C-terminally biotinylated peptides of *C. elegans* histone H3 (amino acids 1–21 and amino acids 14–34, unmodified or with mono/di/trimethylated lysine, for H3K9me or H3K27me, respectively) were chemically synthesized (SciLight Biotechnology, Beijing, China) and used for the pull-down assay. The peptides were coupled to High-Capacity Streptavidin Agarose Resin (Thermo Scientific) according to the manufacturer's instructions. Purified GST fusion proteins (3.7 μM) were incubated with the peptide-bead slurry in binding buffer (25 mM Tris-HCl, pH 7.5, 250 mM NaCl, 5% glycerol, 0.1% Triton X−100) for 1 h at 4 °C on a rotator. After washing five times with binding buffer, bound proteins were released from the beads and resolved by SDS–PAGE followed by Coomassie blue staining.

### Forward genetic screening

Forward genetic screening was conducted as previously described[24]. Briefly, to identify factors regulating CEC-5, we chemically mutagenized GFP::CEC-5 strain by ethyl methanesulfonate (EMS), followed by a clonal screening. The F2 progeny worms were visualized under fluorescent microscope at adult stage. One mutant that disrupted the nuclear puncta formation of CEC-5 was isolated from one thousand haploid genomes. CEC-5(R124C) was identified by genome resequencing.

### RNA isolation

Synchronized adult worms were incubated with TRIzol reagent followed by 4 quick liquid nitrogen freeze-thaw cycles, isopropanol precipitation, and DNaseI digestion (Qiagen).

### qRT−PCR

cDNAs were generated from the RNA using HiScript III RT SuperMix for qPCR (Vazyme), which includes Random primers/Oligo(dT)20VN primer mix for reverse transcription. qPCR was performed using a MyIQ2 real-time PCR system (Bio-Rad) with AceQ SYBR Green Master mix (Vazyme). The primers used in qRT-PCR are listed in Supplementary Table 11.

### Lifespan assay

Lifespan assays were performed at 20°C. Worm populations were synchronized by placing young adult worms on NGM plates seeded with the *E. coli* strain OP50 for 4–6 h and then removed. The hatching day was counted as day one for all lifespan measurements. Worms were transferred every other day to new plates to eliminate confounding progeny. Animals were scored as alive or dead every day. Worms were scored as dead if they did not respond to repeated prods with a platinum pick. Worms were censored if they crawled off the plate or died from vulval bursting and bagging.

### Statistics and reproducibility

For each microscopy result in Figs. 4a−c and 7b; Supplementary Figs. 1; 18a, b; 19; 20f, independent experiments were performed 5 times. For pull-down assay in Fig. 6c−f, independent experiments were performed 3 times.

Besides, the mean and standard error of mean of the results are presented in bar graphs with error bars. All other experiments were conducted with independent *C. elegans* animals for the indicated number (N) of times. Statistical analysis was performed with the one- or two-tailed Student's *t* test or unpaired Wilcoxon test as indicated.

### Reporting summary

Further information on research design is available in the Nature Portfolio Reporting Summary linked to this article.

## Data availability

The data that support this study are available from the corresponding author upon reasonable request. All the high throughput data generated by this work were deposited onto the Genome Sequence Archive in the National Genomics Data Center, China National Center for Bioinformation/Beijing Institute of Genomics, Chinese Academy of Sciences (GSA) with accession code CRA009179). ChIP-seq datasets of histone modifications, epigenetic factors, and reported chromodomain proteins in *C. elegans* were downloaded from the NCBI GEO or ENCODE databases. The datasets used in this study are listed in Supplementary Data 1. Source data are provided with this paper.

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

## Acknowledgements

We are grateful to the members of the Guang lab for their comments. We are grateful to the International *C. elegans* Gene Knockout Consortium and the National Bioresource Project for providing the strains. Some strains were provided by the CGC, which is funded by the NIH Office of Research Infrastructure Programs (P40 OD010440). This work was supported by grants from the Strategic Priority Research Program of the Chinese Academy of Sciences (XDB39010600), the National Key R&D Program of China (2022YFA1302700, 2019YFA0802600), the National Natural Science Foundation of China (32230016, 91940303, 32270583, 32070619, 31870812, 31871300 and 31900434). This study was supported in part by the Fundamental Research Funds for the Central Universities.

## Author contributions

X.H.: Conceptualization, Data curation, Software, Formal analysis, Visualization, Investigation, Methodology, Writing - Original draft. M.X.: Conceptualization, Data curation, Visualization, Investigation, Methodology, Resources. C.Z.: Conceptualization, Investigation, Validation, Data curation, Methodology. J.G.: Software, Formal analysis, Visualization. M.L.: Investigation, Methodology, Resources. X.C.: Conceptualization, Validation. C.S.: Data curation, Validation. B.N.: Conceptualization. J.Z.: Conceptualization. Y.Z.: Conceptualization, Supervision. S.G.: Conceptualization, Supervision, Project administration, Writing - Review & Editing. X.F.: Conceptualization, Supervision, Project administration, Writing - Review & Editing.

## Competing interests

The authors declare no competing interests.

## Additional information

**Peer review information** : *Nature Communications* thanks the anonymous reviewers for their contribution to the peer review of this work. Peer reviewer reports are available.

