## [Peer Review File · Nature Communications]

Systematic characterization of chromodomain proteins reveals an H3K9me1/2 reader regulating aging in *C. elegans*REVIEWER COMMENTS

Reviewer #1 (Remarks to the Author):

Summary

In their manuscript, "Systematic characterization of chromodomain proteins reveals an H3K9me1/2 reader regulating aging in *C. elegans*", Hou, Xu, Zhu and colleagues study a number of chromodomain-containing proteins by characterizing their genome-wide localization and spatial expression patterns. They also perform a directed RNAi screen (and forward genetic screen in the case of CEC-5) to identify genes regulating the sub-nuclear localization of several of the chromodomain proteins. For CEC-5, they identify a potential role for sumoylation, the H3K9me1/2 methyltransferase *met-2*, and residue Arg124 in the CEC-5 chromodomain in the formation of the CEC-5 nuclear puncta. CEC-5 is explored in more detail, which reveals a relative depletion from heterochromatic arms in *met-2* mutants, as well as a reduced lifespan in CEC-5 (and *met-2*) mutants.

A strength of this work is that the library of strains with GFP-tagged chromodomain proteins, the ChIP-seq datasets and the expression patterns are resources that will be of interest and use to the research community. The broad genome-wide comparisons and the correlation network analysis of the genomic binding profiles of the chromodomain proteins along with histone modifications or other chromatin-associated proteins are also strengths.

Since this work is an initial broad survey that includes several uncharacterized genes, it is not unexpected that there is limited mechanistic detail. However, as outlined below, there are several places where more detailed analyses, explanations, or data presentation are required to clarify the findings. In addition, there are inconsistencies between ChIP-seq datasets for the same factor that must be addressed.

Major Points

1. Analysis of the expression patterns of the GFP-tagged proteins is an important result. Images for embryos and germline are shown, but not images of other tissues (where expression is indicated by a "+" in Fig. 1C). Representative microscopy images for the somatic cell types (e.g. neurons, etc.) should be included. In addition, labels or diagrams to accompany the germline images should be included in Fig. S1.
2. The effect of loss of *met-2* and/or *set-25* on the genomic association of CEC-5 is a key result (Fig. 5). The heatmaps (D-E) in the mutant backgrounds appear to show the peaks that are called in the mutant background. However, it is not clear what is happening at all of the peaks normally called in wild-type. To address this question, it would be informative to show all the peaks that are called in the wild-type background and plot the ChIP enrichment for wild-type as well as the mutant backgrounds in parallel (one set of heatmaps with the same rows across, including the histone modifications as well).
3. Related to point 2 above, for the CEC-5 R124C mutant (Fig. 7E), the enrichment of the mutant protein over the peaks that are normally present in wild-type animals would be informative here. An additional question to be addressed is how do the R124C mutation and *met-2* mutation compare in terms of their effect on CEC-5 occupancy?
4. There are major discrepancies among some of the ChIP-seq datasets for the same factor – e.g. three CEC-7 datasets, the two LET-418 datasets – in terms of genome-wide correlations, histone mark co-occupancy (e.g. in Fig. 2E,F, Fig. 3). The authors should discuss this – is there a biological or technical reason? To help clarify this point, an additional suggestion is to evaluate the quality of the ChIP-seq data – e.g. using FRiP and cross-correlation analysis, and to evaluate the 'replicates' using

the IDR method.

5. In Fig. 4D, the RNAi/mutant screen was done on 9 GFP-tagged chromodomains. However, only the 3 hits for CEC-5 are presented. It is therefore unclear what the results were for the other targets. This should be explained, and if any phenotypes were observed, they should be included (e.g. in a supplemental table). In addition, were there any other modifiers identified for CEC-5?

6. The presentation of the (green) peak heatmaps should be improved. First, the description and labelling throughout the figures (e.g. Fig 2 A-D) should include the unit of the colour scale, the number of rows, and what the rows represent (e.g. is this all the called peaks?). Second, please clarify why some heatmaps are divided into clusters, and how the number of clusters was chosen.

7. In Fig. 4F, when comparing the *uba-2* and *ubc-9* RNAi animals with the *met-2* mutants, there appear to be some differences in the pattern of disrupted GFP::CEC-5 (e.g. one large and 1-2 small puncta in the *uba-2/ubc-9* RNAi, vs no puncta in *met-2* mutants). These potentially different patterns should be discussed.

8. For the lifespan assay in Fig7F, the average lifespans (plus/minus error) of each genotype should be calculated and statistics should be used to determine whether the mutant animal lifespans are significantly shorter than wild-type.

9. For the genes studied here that have been more functionally characterized in the literature (e.g. *mrg-1*, *let-418*), the manuscript should mention their roles in more detail and cite the relevant literature.

10. Methods describing the following should be included:

- how was the phylogenetic tree (Fig1A) generated?
- how were the categories of "graded histone modification occupied" (Fig2E, FigS3B) determined?
- how were genomic regions comprising euchromatin/heterochromatin and arm/center determined? (Fig2F-G, FigS2A)
- how was the forward mutagenesis screen carried out

11. An accession number for the deposited sequencing data should be included.

Minor Points

1. For the distribution percentiles of genomic features (promoters, exons, etc.) (Fig2F, FigS4A), it would be helpful to include the overall distribution in the genome (or the indicated sub-region of the genome) as a point of comparison.

2. Labels for the factor being ChIP-ed should be clear and consistent throughout. Note that in Fig. 3A and S4B, two datasets are labelled LET-418(A). In addition, for proteins with previous other data, the "GFP" is included for the current data. (e.g. CHD-3 GFP, CHD-3). It would make sense to include the 'GFP' for other new dataset labels (e.g. GFP-CEC-5 instead of CEC-5). The figure legends and/or labels should also make it clear which datasets are new and which are from previous work, and citations for the previous work should be included in the manuscript text. In some figures, some of the labels are in different colours – the meaning of these colours should be explained.

3. Numbers of animals (not just N>50) and number of biological replicates should be reported for scored assays.

4. In Fig4E & Fig7A, there appears to also be GFP in the nucleoplasm. The authors should consider including this in the cartoon diagram.

5. In FigS7E, ChIP-seq signal is shown across the sequence encoding rRNAs. Please clarify why there is zero signal in the input across parts of the window.

6. Line193 states that 11 genes were successfully targeted, but there appear to be 12 in Fig1C and Table S5.

7. Line218 states that "44% of gene sequences are localized on chromosome arms". Please clarify what "gene sequences" refers to in this context.

8. Line440 - Since binding affinity was not measured, this statement should be rephrased

9. Line 698 - refers to a method of "Blondel et al" but no reference is given.

Reviewer #2 (Remarks to the Author):

In the manuscript "Systematic characterization of chromodomain proteins reveals an H3K9me1/2 reader regulating aging in *C. elegans*" Hou, Xu, Zhu and colleagues provide an overview and toolset for the analysis of chromodomain containing proteins in *C. elegans*. Chromodomain containing proteins are important components of chromatin structure and mediators of chromatin function. A systematic understanding of their dynamics and function would therefore be of great importance to the field. In its current state the manuscript provides interesting datasets and tools (in the form of the collection of tagged chromodomain proteins), however there are certain parts of the analysis that need to be improved and expanded before it can be truly considered to be a systematic study of chromodomain proteins.

My major points are:

1) The description of chromodomain localization is not systematic and needs to be a) expanded to provide a detailed and consistent expression/localization pattern from at least germline and embryos for all tagged chromodomain proteins b) the scheme of expression must be replaced with an actual quantitative measurement that supports the collection of representative images.

In more detail:

Figure S1A: the authors state their observation of how the tagged CD proteins are expressed in the germline, however from the example images germline localization is very much unclear. Please mark the germline and give examples plus an actual quantification of intensities in germline stages / zones. As presented right now there are apparent discrepancies between the scheme in Fig. 1d and Fig. S1a,b. For example CEC-2 and CEC-10 appear as if they are not expressed in early embryos and CEC-2 doesn't appear to be expressed in the gonads, while CEC-10 probably is at least in some stages. This conflicts with the scheme in Fig. 1d. Also in the text the authors describe CEC-7 as "exhibited a dramatic reduction in expression at the transition zone but was re-expressed in oocytes" (Page 7 Line 201-202). The corresponding image shown in Fig. S1a shows a re-expression somewhere around pachytene. Please also include CEC-8 in the collection of germline images and the quantification.

Figure 4 This is not a systematic analysis It is important to know from which location in the germline images have been taken and both images should be comparable for all Chromodomain proteins and stages. E.g. For the germline show localization of mitotic zone cells, meiotic cells, and oocytes. For embryos show an early and a late embryo. Also please provide both an overview and an image focusing on 1-4 nuclei.

In addition the authors show two very different localization pattern for CEC-5 and none for UAD-2 despite it being mentioned in the text.

2) The analysis of the ChIP-seq coverage is hard to interpret from the Figures presented and certain

analysis are missing (see below)

In more detail:

- Page 7-8 For the ChIP-seq experiments please provide some form of quality control. For example please correlate your dataset generated with the GFP antibody for LET-418 with the published LET-418 dataset. Also please provide MA plots so that readers can estimate the range of enrichment and background. While it is common practice for *C. elegans* literature to show whole chromosomes the resolution of the tracks shown in Fig 1e is very low. It would be more informative to show a representative autosome and the X-chromosome in the main Figure and move the remaining chromosomes to supplementary.
- Regarding peak calling please provide an overview of how many peaks were called and the peak sizes. In addition please provide representative genome tracks showing examples of peaks with the aligned data. Especially heterochromatic proteins often show a broad distribution and for example peak calling can be very inaccurate for example for histone marks like H3K9me2/me3 because of that. Can the authors explain why they didn't choose the broad peak function integrated in MACS2?
- Page 8-9 (Line 238-247) please provide correlation coefficient for every correlation mentioned.
- For Fig. 2a-d and Fig S2B-E and 3A: While the heatmaps are a information rich way of presenting the data, they are hard to translate into numbers and quantitative statements of enrichments. The additional provided correlation coefficients are in some way helpful, however for example UAD-2, CHD-7, SET-31 appear to cover the mutual exclusive H3K4me3 marked regions and H3K27me3 / H3K9me2/me3 regions. When analyzed together this weakens a overall correlation coefficient, despite clear overlap. It is therefore important to separate these and provide numbers (% of chromodomain peaks cover a certain PTM and vice versa).
- One strategy could be: From Figure 2f I conclude that the majority of peaks cover a gene and a number of peaks cover intergenic sequences and repeats. In addition to the Fig.2a-d it would be informative to provide either heatmaps, or for the sake of clarity average coverage plots showing the coverage of the different chromodomain proteins at actively transcribed and silent genes (based on RNA expression, and Histone PTMs). For example centered around the transcription start site. There is no need to complicate things with a separation between "arms and center" as these are super broad regions and by no means a substitute for the proper correlation with histone PTMs. The analogous plots could be made with repetitive elements (e.g. split between DNA transposons, RNA transposons and simple repeats/satellite sequences). This would provide more information about how the chromodomain protein cover genes and or promoter and their relationship to the histone PTMs. In addition the authors should provide the % of chromodomain peaks cover a certain PTM and vice versa, as mentioned earlier.
- Page 9: The authors conclude here that "Most chromodomain proteins were associated with both heterochromatin and euchromatin", however ChIP-seq experiments were performed in whole animals and it is therefore likely that especially at genes that are differentially expressed in different tissues the authors look at a mixture of repressed and active genes and it is not clear whether the signal from the Chromodomain proteins reflects the binding to one or the other. As informative as their datasets are, this is a serious limitation and prohibits in my eyes conclusions that go beyond "protein x is enriched at chromatin state Y".
- Page 10 Line 272: "UAD-2 was highly enriched in piRNA genes" Indeed more UAD-2 peaks appear to cover piRNA genes compared to the other chromodomain proteins. However from this graph it is not clear whether UAD-2 is enriched at piRNA genes compared to their occurrence in the genome (given the lower number of piRNA genes compared to coding genes that is very likely and the plot likely).
- Page 10 Line 279-280: "UAD-2 was not enriched in any specific GO (Gene Ontology) terms, which is

consistent with its preferential binding to piRNA genes” This sentence does not make sense to me. According to Figure S4a the majority of UAD-2 peaks cover protein coding genes. Therefore, it is only a preferential binding to piRNA loci when compared to the other chromodomain proteins.

- In addition to the correlation analysis and the coverage of the individual chromodomain proteins at genes and repeats it would be informative to plot the overlap of the chromodomain proteins at genes/repeats and then ask what is the chromatin state of the genes/repeats that show a certain combination of chromodomain proteins.
- Page 14 (Line 411-424): Please discuss that because of the lack of a spike in control, one can not say whether the appearance of novel CEC-5 peaks in a met-2 mutant is due to an increased binding of CEC-5 to these regions, or whether there is a general loss of chromatin association of CEC-5 and the signal observed in the ChIP-seq is noise amplified due to PCR and normalization of the data.

3) The link between CEC-5 and the nucleolus is interesting, especially in the context of the aging phenotype, however it remains weak. It would be strengthened if the authors could quantify rRNA expression in WT, met-2 and cec-5 mutants. In flies loss of Su(var)3-9 results in the fragmentation of nucleoli and rDNA instability (Peng et al., 2007 and 2009). The authors could check if this is also the case for CEC-5 mutant embryos (e.g. using the FIB-1::GFP strain).

Minor points:

I would ask the authors to change the colors of the c elegans proteins in Fig. 1a to make it more noticeable. Also please distinguish human and C. elegans proteins only by color and remove the individual text to make it more accessible.

Page 11 Line 309: please cite Garrigues et al., 2015 for HPL-2; Padeken et al., 2019 and 2021 for MET-2 and LIN-61 and Koester-Eiserfunke et al., 2011 for LIN-61

Page 15 Line 439-441: “Surprisingly, CEC-5 bound both unmethylated (me0) and methylated (me1/2/3) H3K9 peptides, although CEC5 CD showed weaker binding affinity to unmethylated (me0) H3K9 peptide (Figs.6C, 6D).” please add a statement that this reflects the binding of only the isolated chromodomain under these invitro conditions. In my eyes the low affinity to H3K9me0 in this assay plus the actual clear invivo dependency on H3K9me1/me2 for binding makes it far more likely that CEC-5 is indeed binding H3K9 when methylated.

Page 15 Line 439-441: “Since the binding of CEC-5 to chromosomes depends on H3K9me1/2 but not H3K9me3 in vivo, we speculated that other amino acid sequences of CEC-5 may modulate the binding specificity of the CEC-5 chromodomain” Please note that H3K9me2 is sufficient for CEC-5 binding. This doesn’t mean that it does not H3K9me3. The H3K9me1/me2 provided by MET-2 is essential for ~80% of all the H3K9me3 catalyzed by SET-25 and in the absence of SET-25 these regions retain H3K9me2. (Padeken et al., 2019).

In the same context please change the title. Given the invitro data and the correct interpretation of the genetic experiment one can not conclude that CEC-5 is a “H3K9me1/2 reader” instead it is a reader of H3K9me1/me2/me3 (or H3K9me).

Page 17 Line 496-497: “We identified a point mutation in the CEC-5 chromodomain critical for its recognition of methyl-lysine of histone 3” Please rephrase: In the data presented in Figure 7 the authors showed that the R124C mutation results in a loss of proper CEC-5 localization. This does not show whether it is due to an inability to bind H3K9me.

REVIEWER COMMENTS

Reviewer #1 (Remarks to the Author):

Summary

In their manuscript, “Systematic characterization of chromodomain proteins reveals an H3K9me1/2 reader regulating aging in C. elegans”, Hou, Xu, Zhu and colleagues study a number of chromodomain-containing proteins by characterizing their genome-wide localization and spatial expression patterns. They also perform a directed RNAi screen (and forward genetic screen in the case of CEC-5) to identify genes regulating the sub-nuclear localization of several of the chromodomain proteins. For CEC-5, they identify a potential role for sumoylation, the H3K9me1/2 methyltransferase met-2, and residue Arg124 in the CEC-5 chromodomain in the formation of the CEC-5 nuclear puncta. CEC-5 is explored in more detail, which reveals a relative depletion from heterochromatic arms in met-2 mutants, as well as a reduced lifespan in CEC-5 (and met-2) mutants.

A strength of this work is that the library of strains with GFP-tagged chromodomain proteins, the ChIP-seq datasets and the expression patterns are resources that will be of interest and use to the research community. The broad genome-wide comparisons and the correlation network analysis of the genomic binding profiles of the chromodomain proteins along with histone modifications or other chromatin-associated proteins are also strengths.

Since this work is an initial broad survey that includes several uncharacterized genes, it is not unexpected that there is limited mechanistic detail. However, as outlined below, there are several places where more detailed analyses, explanations, or data presentation are required to clarify the findings. In addition, there are inconsistencies between ChIP-seq datasets for the same factor that must be addressed.

Major Points

1. Analysis of the expression patterns of the GFP-tagged proteins is an important result. Images for embryos and germline are shown, but not images of other tissues (where expression is indicated by a “+” in Fig. 1C). Representative microscopy images for the somatic cell types (e.g. neurons, etc.) should be included. In addition, labels or diagrams to accompany the germline images should be included in Fig. S1.

RESPONSE: Thanks very much for the suggestions.

In the new Fig. S1, microscopy images for GFP-tagged chromodomain proteins in somatic cells (neuron, intestine, and epidermis) are included. In the new Fig. S2A, germline is indicated by white dashed line. Mitotic region, transition zone, pachytene, diplotene, and oocyte are labeled.

2. The effect of loss of *met-2* and/or *set-25* on the genomic association of CEC-5 is a key result (Fig. 5). The heatmaps (D-E) in the mutant backgrounds appear to show the peaks that are called in the mutant background. However, it is not clear what is happening at all of the peaks normally called in wild-type. To address this question, it would be informative to show all the peaks that are called in the wild-type background and plot the ChIP enrichment for wild-type as well as the mutant backgrounds in parallel (one set of heatmaps with the same rows across, including the histone modifications as well).

RESPONSE: Thanks very much for the suggestions.

In the new Fig. 5F, we compared the ChIP enrichment of CEC-5 for wild-type and mutant animals (*met-2* single mutant, *set-25* single mutant and *met-2;set-25* double mutant) at all of the peaks normally called in wild-type animals. 779 out of the 3558 targets showed dramatically reduced CEC-5 binding in *met-2* single mutant and *met-2;set-25* double mutant, but not in *set-25* single mutant. In addition, the reduced CEC-5 binding targets were occupied by H3K9me2. These data suggested that MET-2 and H3K9me1/2 were required for the proper association of CEC-5 with chromatin.

The text was revised accordingly (lines 481-488).

3. Related to point 2 above, for the CEC-5 R124C mutant (Fig. 7E), the enrichment of the mutant protein over the peaks that are normally present in wild-type animals would be informative here. An additional question to be addressed is how do the R124C mutation and *met-2* mutation compare in terms of their effect on CEC-5 occupancy?

RESPONSE: Thanks very much for the suggestions.

In the new Fig. 7F, we compared the ChIP enrichment of CEC-5 for wild-type and CEC-5(R124C) animals at all of the peaks normally called in wild-type animals. 746 out of the 3558 targets showed dramatically reduced CEC-5 binding in *R124C*. The reduced CEC-5 binding targets were also occupied by H3K9me2.

In the new Fig. 7G, to compare CEC-5 occupancy in *R124C* mutation and *met-2* mutation, we overlapped the reduced CEC-5 targets in *met-2* and *R124C* mutants. Strikingly, 745 out of 746 reduced *R124C* targets were overlapped with those in *met-2* mutant. Taken together, these results suggested that the Arg124 residue in chromodomain played important roles for CEC-5 to recognize H3K9me1/2.

The text was revised accordingly (lines 537-544).

4. There are major discrepancies among some of the ChIP-seq datasets for the same factor – e.g. three CEC-7 datasets, the two *LET-418* datasets – in terms of genome-wide correlations, histone mark co-occupancy (e.g. in Fig. 2E,F, Fig. 3). The authors should discuss this – is there a

biological or technical reason? To help clarify this point, an additional suggestion is to evaluate the quality of the ChIP-seq data – e.g. using FRiP and cross-correlation analysis, and to evaluate the ‘replicates’ using the IDR method.

RESPONSE: Thanks very much for the suggestions. We performed quality control of all the chromodomain protein ChIP-seq datasets used in this work as suggested.

Fraction of Reads in Peaks (FRiP) of each sample was higher than 0.01 (new Fig. S4A, new Table S5). Then we performed cross-correlation analysis. Most of the samples showed clear fragment-length peak (new Fig. S4B). The normalized ratio between the fragment-length cross-correlation peak and the background cross-correlation (normalized strand coefficient, NSC) and the ratio between the fragment-length peak and the read-length peak (relative strand correlation, RSC) of each chromodomain proteins were calculated (new Table S5). Most of RSC values were higher than 0.8. However, for unknown reasons, the NSC values in 15 out of 18 samples were lower than 1.05. MA-plots were included to show the enrichment of each chromodomain protein versus input (new Fig. S3).

We performed correlation (new Fig. S5A) and IDR (new Figs. S5B-F) analysis for CHD-3, LET-418 and CEC-7, respectively. CHD-3 GFP(A) and published CHD-3 SDQ3907(A) were positively correlated (Pearson correlation coefficient $r=0.385$). LET-418 GFP(A) and published LET-418(A) were positively correlated ($r=0.389$). CEC-7 GFP(A) was weakly positively correlated with CEC-7 SDQ5413(A) and CEC-7 SDQ5421(A) ($r=0.241$ and $r=0.255$, respectively), whereas CEC-7 SDQ5413(A) was strongly positively correlated with CEC-7 SDQ5421(A) ($r=0.502$) (new Fig. S5A). A two-sided t -test was performed for each correlation analysis and all $P < 2.2 \times 10^{-16}$.

For IDR analysis (new Figs. S5B-F), CHD-3 GFP(A) and published CHD-3 SDQ3907(A) displayed 15.8% peaks (383 out of 2430) passing IDR cutoff 0.05. LET-418 GFP(A) and published LET-418(A) had 2% peaks (50 out of 2474) passing the cutoff. CEC-7 GFP(A) shared 76.6% (85 out of 111) and 18.2% (87 out of 477) significant peaks with CEC-7 SDQ5413(A) and SDQ5421(A), respectively. The published CEC-7 samples (CEC-7 SDQ5413(A) and CEC-7 SDQ5421(A)) had 1% peaks (41 out of 4005) passing IDR cutoff 0.05.

The method section was revised to include the information (lines 817-846).

In this work, we compared chromodomain protein ChIP-seq datasets obtained using GFP antibody (ab290) in our experiments and those previously published CHD-3, LET-418, and CEC-7 datasets. CHD-3 GFP(A) and published CHD-3 SDQ3907(A) both showed that CHD-3 associated with H3K4me3 (Figs. 2F, S9B and 3A-B).

Both LET-418 GFP(A) and LET-418(A) ChIP-seq datasets showed that LET-418 was related to H3K4me3 (Figs. 2F, S9B and 3A-B). However, LET-418(A) but not

LET-418 GFP(A) showed that LET-418 was also correlated with H3K9me2 (Figs. 2F, 3B). Previous work (the LET-418(A) data) revealed that LET-418, HPL-2, LIN-13, LIN-61, and MET-2 were all associated with H3K9me2 and repetitive elements whereas LET-418 exhibited generally lower enrichment on repetitive sequences than the other factors (Alicia N McMurchy, et al. 2017 Elife). We speculated LET-418 may recognize both H3K4me3 and H3K9me2. The discrepancy might be caused by different antibodies used (ab290 vs. Q3861) or worm culture conditions (OP50 vs. HB101 *E. coli*).

CEC-7 GFP(A) suggested that CEC-7 may associate with H3K4me3. Although CEC-7 SDQ5421(A) revealed weak association with H3K4me3 at chromosome centers, CEC-7 SDQ5413(A) and CEC-7 SDQ5421(A) showed that CEC-7 may mainly associate with H3K36me3 and H3K79me3. The CEC-7 SDQ5413(A) and CEC-7 SDQ5421(A) datasets were downloaded from modENCODE project, using antibody SDQ5413 and SDQ5421, respectively. The reason of discrepancy is unclear.

The discussion section was revised to include the information (lines 691-713).

5. In Fig. 4D, the RNAi/mutant screen was done on 9 GFP-tagged chromodomains. However, only the 3 hits for CEC-5 are presented. It is therefore unclear what the results were for the other targets. This should be explained, and if any phenotypes were observed, they should be included (e.g. in a supplemental table). In addition, were there any other modifiers identified for CEC-5?

RESPONSE: Thanks very much for the comments.

For UAD-2, we have previously identified a number of epigenetic factors regulation the subcellular localization of UAD-2 and piRNA transcription (Huang, 2021 PNAS).

Here, we showed that 3 hits for CEC-5 localization (Fig. 4F). We did not notice pronounced subcellular change of other chromodomain proteins by RNAi knocking down the epigenetic factors. Right now, we are still conducting forward genetic screening to search for factors regulating the subcellular localization of CEC-5.

The results section was revised to include this information (lines 436-440, 445-447).

6. The presentation of the (green) peak heatmaps should be improved. First, the description and labelling throughout the figures (e.g. Fig 2 A-D) should include the unit of the colour scale, the number of rows, and what the rows represent (e.g. is this all the called peaks?). Second, please clarify why some heatmaps are divided into clusters, and how the number of clusters was chosen.

RESPONSE: Thanks very much for the suggestions.

We have revised the peak heatmaps by including the unit of color scale, the number of rows, and what the rows represent in labelling and figure legends (Figs. 2B-E, S8D-G, and S9A).

To identify heterogeneity of chromatin states of chromodomain protein binding targets, heatmap of each chromodomain protein was clustered. Parameter --kmeans was used to define the number of clusters. For each chromodomain protein, --kmeans 1, 2, and 3 were all applied. Plots displaying no heterogeneity of chromatin states within each cluster were selected. A plot with minimum kmeans number for each chromodomain protein from the selected plots was shown.

The methods section was revised to include this information (lines 877-884).

7. In Fig. 4F, when comparing the uba-2 and ubc-9 RNAi animals with the met-2 mutants, there appear to be some differences in the pattern of disrupted GFP::CEC-5 (e.g. one large and 1-2 small puncta in the uba-2/ubc-9 RNAi, vs no puncta in met-2 mutants). These potentially different patterns should be discussed.

RESPONSE: Thanks very much for the comments.

There appeared to be different subcellular localization patterns of CEC-5 upon *uba-2* and *ubc-9* RNAi compared to that in *met-2* animals. One large and 1-2 small puncta remained in the *uba-2/ubc-9* RNAi, but not in *met-2* animals. The mechanism is unclear. It is possible that SUMOylation might have different effects on CEC-5 occupancy at certain genomic loci compared to H3K9me1/2 marks.

The results section was revised to include this information (lines 468-472).

8. For the lifespan assay in Fig7F, the average lifespans (plus/minus error) of each genotype should be calculated and statistics should be used to determine whether the mutant animal lifespans are significantly shorter than wild-type.

RESPONSE: Thanks very much for the suggestions.

In the new Fig. 7I, the average lifespans of each genotype were calculated and statistics were shown. Another two replicates of lifespan experiments were performed and the results were included (new Figs. S21A-D).

9. For the genes studied here that have been more functionally characterized in the literature (e.g. mrg-1, let-418), the manuscript should mention their roles in more detail and cite the relevant literature.

RESPONSE: Thanks very much for the suggestions.

The text was revised accordingly to include previous works (lines 116-117, 121-127, and 136-139).

10. *Methods describing the following should be included:*

-how was the phylogenetic tree (Fig1A) generated?

RESPONSE: Thanks very much for the suggestions.

Protein sequences were obtained from Wormbase.org and UniProt.org. There are 26 chromodomain protein sequences in *C. elegans* (different isoforms were included), and 38 sequences in human (only one isoform of the same protein was adopted). Protein sequences were aligned using ClustalW with MEGA-X (Kumar et al., Mol. Biol. Evol. 2018). The phylogenetic tree was inferred by using the Neighbor-Joining method. Poisson model was chosen as the substitution model and the Uniform Rates was used to model evolutionary rate differences among sites. Bootstrap analysis was based on 1000 replicates using the Neighbor-Joining method.

The method section was revised to include this information (lines 737-745).

-how were the categories of “graded histone modification occupied” (Fig2E, FigS3B) determined?

RESPONSE : “Graded histone modification occupied” for a pair of chromodomain protein and histone modification fell into 3 categories: grade 3 = “prominent”, grade 2 = “detectable”, and grade 1 = “weak”. Grade 3 was defined as pairs meeting all of the 3 criteria: 1. “Pearson correlation coefficient (r) ≥ 0.300 ”. 2. “ $\geq 30\%$ of significant peaks occupied by a certain histone modification (using IntervalStats software package)”. 3. “Overlap detected by heatmap analysis”. Grade 2 meets 2 out of the 3 criteria and grade 1 meets any of the 3 criteria.

For example, on chromosome arms, Pearson correlation coefficient of CEC-5 GFP(A) and H3K9me2, $r = 0.333$ (Fig. S10); 44.3% peaks of CEC-5 were significantly occupied by H3K9me2 (Fig. 2A and Table S7); Heatmap analysis found 1585 out of 1854 targets of CEC-5 were covered by H3K9me2 (Fig. 2B). Therefore, the association of CEC-5 and H3K9me2 on chromosome arms was classified as “grade 3 = prominent” (Fig. 2F and Table S8).

The method section was revised to include this information (lines 900-911).

-how were genomic regions comprising euchromatin/heterochromatin and arm/center determined? (Fig2F-G, FigS2A)

RESPONSE : Chromosome arms and centers were determined as reported (Barnes, et al. 1995 Genetics; Rockman and Kruglyak, 2009 PLoS Genet; Tao Liu, et al. 2011 Genome Res). Briefly, recombination rates and H3K9 methylation enrichment were used to estimate the physical boundaries between the arms and central region of each chromosome. Chromosome arms were characterized by high levels of recombination rate and H3K9 methylation. Coordinates of arms and center regions of each chromosome were shown in Table S6.

Heterochromatin was defined by occupancy of H3K9me2, H3K9me3, and H3K27me3, and euchromatin was determined by covering of H3K4me3, H3K36me3, and H3K79me3 (Barski et al. 2007 Cell; Muller J, Verrijzer P. 2009 Curr Opin Genet Dev; Tao Liu, et al. 2011 Genome Res; Talbert, P. B et al. 2019 Nature Reviews Genetics; Millan-Zambrano, G. et al. 2022 Nature Reviews Genetics). Association between chromodomain proteins and histone modifications was detected by correlation, heatmap, and peak overlapping analysis.

The method section was revised to include the information (lines 849-862).

-how was the forward mutagenesis screen carried out

RESPONSE : Forward genetic screening was conducted as previously described (Huang, 2021 PNAS). Briefly, to identify factors regulating CEC-5, we chemically mutagenized GFP::CEC-5 strain by ethyl methanesulfonate (EMS), followed by a clonal screen. The F2 progeny worms were visualized under fluorescent microscope at adult stage. One mutant that disrupted the nuclear puncta formation of CEC-5 was isolated from one thousand haploid genomes. CEC-5(R124C) was identified by genome re-sequencing.

The method section was revised to include the information (lines 981-988).

11. An accession number for the deposited sequencing data should be included.

RESPONSE: All the high throughput data were deposited onto the Genome Sequence Archive in the National Genomics Data Center, China National Center for Bioinformation/Beijing Institute of Genomics, Chinese Academy of Sciences (GSA: CRA009179), which are publicly accessible at <https://bigd.big.ac.cn/gsa/browse/CRA009179>.

Minor Points

1. For the distribution percentiles of genomic features (promoters, exons, etc.) (Fig2F, FigS4A), it would be helpful to include the overall distribution in the genome (or the indicated sub-region of the genome) as a point of comparison.

RESPONSE: Thanks very much for the suggestions.

The distribution percentiles of genomic features in the genome (or the indicated sub-region of the genome) were included in Fig. 2G and Fig. S14A.

2. Labels for the factor being ChIP-ed should be clear and consistent throughout. Note that in Fig. 3A and S4B, two datasets are labelled LET-418(A). In addition, for proteins with previous other data, the “GFP” is included for the current data. (e.g. CHD-3 GFP, CHD-3). It would make sense to include the ‘GFP’ for other new dataset labels (e.g. GFP-CEC-5 instead of CEC-5). The figure legends and/or labels should also make it clear which datasets are new and which are from previous work, and citations for the previous work should be included in the manuscript text. In some figures, some of the labels are in different colours – the meaning of these colours should be explained.

RESPONSE: Thanks very much for the suggestions.

We have corrected the labels and made them consistent throughout. All the ChIP-seq datasets newly generated in this work by GFP antibody were labelled as “Gene name GFP(Stage)” (e.g. CHD-1 GFP(A)). Previously published datasets were labelled as “Gene name(Stage)” (e.g. HPL-2(L3)).

Citations for the previous work were included. The text was revised accordingly (lines 224-225).

In Fig. 3A, colors indicate different samples. chromodomain protein samples generated in this work, green; chromodomain protein samples downloaded, red; histone modification and epigenetic regulators, black. The figure legend was revised to include the information (Fig. 3A).

3. Numbers of animals (not just $N > 50$) and number of biological replicates should be reported for scored assays.

RESPONSE: Thanks very much for the suggestion.

Numbers of animals were labeled in Figs. 4G and S20B. For lifespan assays, numbers of animals and biological replicates were included in Fig. 7H and the two replicates (new Figs. S21A-B).

4. In Fig4E & Fig7A, there appears to also be GFP in the nucleoplasm. The authors should consider

including this in the cartoon diagram.

RESPONSE: The cartoon diagrams of Figs. 4E and 7A were corrected as suggested.

5. In FigS7E, ChIP-seq signal is shown across the sequence encoding rRNAs. Please clarify why there is zero signal in the input across parts of the window.

RESPONSE: Thanks very much for the comments.

We plotted coverage plots showing ChIP-seq signals across the sequence encoding rRNAs with Deeptools subcommand plotProfile (version 3.4.3). Regions -0.5kb upstream and 0.5kb downstream the rDNA sequences were included (Fig. S20C).

We checked IGV tracks showing ChIP-seq signals on the right arm of chr I in bigwig files of CEC-5 GFP(A) (red) and input (blue). Data range was settled as 0-500 (See below). Rare signals were detected upstream or downstream the rDNA sequences.

Then we changed the data range to 0-7.5 (See below). Weak signals were detected in both CEC-5 GFP(A) (red) and input (blue) files outside the rDNA gene body.

We speculated the signal was too weak to be detected upstream or downstream the rDNA sequence in Fig. S20C may be the result of biased PCR amplification in ChIP-seq.

6. Line193 states that 11 genes were successfully targeted, but there appear to be 12 in Fig1C and Table S5.

RESPONSE: Sorry for the typo. Of the 21 chromodomain genes in *C. elegans*, we successfully targeted 12 of them with a GFP-3xFLAG fluorescent tag.

The text was revised (line 202).

7. Line218 states that “44% of gene sequences are localized on chromosome arms”. Please clarify what “gene sequences” refers to in this context.

RESPONSE: “Gene sequences” was referred to “genomic regions”. The text was corrected accordingly (line 233). The genomic regions were defined as chromosome arms and centers based on previous works.

Briefly, recombination rates and H3K9 methylation enrichment were used to estimate the physical boundaries between the arms and central region of each chromosome. Chromosome arms were characterized by high levels of recombination rate and H3K9 methylation (Barnes, et al. 1995 Genetics; Rockman and Kruglyak, 2009 PLoS Genet; Tao Liu, et al. 2011 Genome Res). Coordinates of arms and center regions of each chromosome were shown in Table S6. The methods section was revised to include the information (lines 849-855).

8. Line440 - Since binding affinity was not measured, this statement should be rephrased

RESPONSE: We agree with the reviewer. The sentence was deleted.

9. Lind 698 – refers to a method of “Blondel et al” but no reference is given.

RESPONSE: Sorry for the mistake. The reference was included.

Reviewer #2 (Remarks to the Author):

In the manuscript “Systematic characterization of chromodomain proteins reveals an H3K9me1/2 reader regulating aging in C. elegans” Hou, Xu, Zhu and colleges provide an overview and toolset for the analysis of chromodomain containing proteins in C. elegans. Chromodomain containing proteins are important components of chromatin structure and mediators of chromatin function. A systematic understanding of their dynamics and function would therefore be of great importance to the field.

In its current state the manuscript provides interesting datasets and tools (in the form of the collection of tagged chromodomain proteins), however there are certain parts of the analysis that

need to be improved and expanded before it can be truly considered to be a systematic study of chromodomain proteins.

My major points are:

1) The description of chromodomain localization is not systematic and needs to be a) expanded to provide a detailed and consistent expression/localization pattern from at least germline and embryos for all tagged chromodomain proteins b) the scheme of expression must be replaced with an actual quantitative measurement that supports the collection of representative images.

In more detail:

Figure S1A: the authors state their observation of how the tagged CD proteins are expressed in the germline, however from the example images germline localization is very much unclear. Please mark the germline and give examples plus an actual quantification of intensities in germline stages / zones. As presented right now there are apparent discrepancies between the scheme in Fig. 1d and Fig. S1a,b. For example CEC-2 and CEC-10 appear as if they are not expressed in early embryos and CEC-2 doesn't appear to be expressed in the gonads, while CEC-10 probably is at least in some stages. This conflicts with the scheme in Fig. 1d. Also in the text the authors describe CEC-7 as "exhibited a dramatic reduction in expression at the transition zone but was re-expressed in oocytes" (Page 7 Line 201-202). The corresponding image shown in Fig. S1a shows a re-expression somewhere around pachytene. Please also include CEC-8 in the collection of germline images and the quantification.

RESPONSE: Thanks for the comments.

We have revised images showing germline localization in Fig. S2A. Germline of each image was indicated by white outlines. Quantification of GFP intensities was performed by using ImageJ (v1.8.0). The relative GFP intensity (relative to the average level of GFP::CEC-5 at mitotic region) of mitotic, transition zone, pachytene, diplotene, and oocyte zones was calculated, respectively. Four independent replicates were used. The information was included in Fig. 1D and methods section (lines 749-753).

We found UAD-2 and SET-31 were expressed in the mitotic and meiotic regions but not in oocytes (Huang, 2021 PNAS). CEC-5 was highly expressed in the whole germline but declined during meiosis. The expression level of CHD-1 also reduced slightly during oogenesis. In contrast, CHD-3, CHD-7, and CEC-2 were not expressed in the mitotic and early meiotic regions but began to be expressed in diplotene cells. Notably, CEC-7 exhibited a dramatic reduction in expression at the transition zone but was re-expressed in late pachytene cells (Figs. 1D, S2A). CEC-8 was not expressed in the germline. Other chromodomain proteins were expressed constantly throughout the germline. Fig. 1D and Fig. S2A were corrected. The text was corrected accordingly (lines 206-215). Germline images of CEC-8 was included and quantified (Figs. 1D and S2A).

Figure 4 This is not a systematic analysis It is important to know from which location in the germline

images have been taken and both images should be comparable for all Chromodomain proteins and stages. E.g. For the germline show localization of mitotic zone cells, meiotic cells, and oocytes. For embryos show an early and a late embryo. Also please provide both an overview and an image focusing on 1-4 nuclei.

In addition the authors show two very different localization pattern for CEC-5 and none for UAD-2 despite it being mentioned in the text.

RESPONSE: Thanks for the suggestions.

We have included a new Fig. S18 to show the localization of all chromodomain proteins in mitotic, meiotic cells and oocytes. Overviews and images focusing on 1-4 nuclei for early and late embryos, respectively, were included in new Fig. S19. Summary of subcellular localization of chromodomain proteins was included in Table S10.

The subcellular localization of CEC-5 diverged in different tissues or developmental stages. In mitotic zone or early meiotic cell, CEC-5 mainly localized to nuclear puncta on chromosomes (Figs. 4B, S18A). In oocytes, CEC-5 was mainly enriched in nucleolus (Fig. S20A). In early embryos, CEC-5 formed nuclear puncta on chromosomes (Fig. S19). However, in late embryos, CEC-5 colocalized with nucleolar marker FIB-1 (Fig. 4C). In addition, CEC-5 constantly localized to nucleolus in somatic cells (Fig. S20A). The text was revised to include the information (lines 411-417).

UAD-2 localized to chromatin and formed nuclear foci in mitotic zone and early meiotic cells (Figs. 4B and new S18A) (Huang, 2021 PNAS). In embryos, UAD-2 colocalized with chromosomes during mitotic metaphase (Fig. S19). The text was revised to include the information (lines 404-406).

2) The analysis of the ChIP-seq coverage is hard to interpret from the Figures presented and certain analysis are missing (see below)

In more detail:

• Page 7-8 For the ChIP-seq experiments please provide some form of quality control. For example please correlate your dataset generated with the GFP antibody for LET-418 with the published LET-418 dataset. Also please provide MA plots so that readers can estimate the range of enrichment and background. While it is common practice for C. elegans literature to show whole chromosomes the resolution of the tracks shown in Fig 1e is very low. It would be more informative to show a representative autosome and the X-chromosome in the main Figure and move the remaining chromosomes to supplementary.

RESPONSE: Thanks for the suggestions. We performed quality control of all chromodomain ChIP-seq datasets used in this work as suggested.

Fraction of Reads in Peaks (FRiP) of each sample was higher than 0.01 (new Fig. S4A, new Table S5). Then we performed cross-correlation analysis. Most of the samples showed clear fragment-length peak (new Fig. S4B). The normalized ratio between the fragment-length cross-correlation peak and the background cross-correlation (normalized strand coefficient, NSC) and the ratio between the fragment-length peak and the read-length peak (relative strand correlation, RSC) of each chromodomain proteins were calculated (new Table S5). Most of RSC values were higher than 0.8. However, for unknown reasons, the NSC values in 15 out of 18 samples were lower than 1.05. MA-plots were included to show the enrichment of each chromodomain protein versus input (new Fig. S3).

We performed correlation (new Fig. S5A) and IDR (new Figs. S5B-F) analysis for CHD-3, LET-418 and CEC-7, respectively. CHD-3 GFP(A) and published CHD-3 SDQ3907(A) were positively correlated (Pearson correlation coefficient $r=0.385$). LET-418 GFP(A) and published LET-418(A) were positively correlated ($r=0.389$). CEC-7 GFP(A) was weakly positively correlated with CEC-7 SDQ5413(A) and CEC-7 SDQ5421(A) ($r=0.241$ and $r=0.255$, respectively), whereas CEC-7 SDQ5413(A) was strongly positively correlated with CEC-7 SDQ5421(A) ($r=0.502$) (new Fig. S5A). A two-sided *t*-test was performed for each correlation analysis and all $P < 2.2 \times 10^{-16}$.

For IDR analysis (new Fig. S5B-F), CHD-3 GFP(A) and published CHD-3 SDQ3907(A) displayed 15.8% peaks (383 out of 2430) passing IDR cutoff 0.05. LET-418 GFP(A) and published LET-418(A) had 2% peaks (50 out of 2474) passing the cutoff. CEC-7 GFP(A) shared 76.6% (85 out of 111) and 18.2% (87 out of 477) significant peaks with CEC-7 SDQ5413(A) and SDQ5421(A), respectively. The published CEC-7 samples (CEC-7 SDQ5413(A) and CEC-7 SDQ5421(A)) had 1% peaks (41 out of 4005) passing IDR cutoff 0.05.

The method section was revised to include the information (lines 817-846).

In this work, we compared chromodomain protein ChIP-seq datasets obtained using GFP antibody (ab290) in our experiments and those previously published CHD-3, LET-418, and CEC-7 datasets. CHD-3 GFP(A) and published CHD-3 SDQ3907(A) both showed that CHD-3 associated with H3K4me3 (Figs. 2F, S9B and 3A-B).

Both LET-418 GFP(A) and LET-418(A) ChIP-seq datasets showed that LET-418 was related to H3K4me3 (Figs. 2F, S9B and 3A-B). However, LET-418(A) but not LET-418 GFP(A) showed that LET-418 was also correlated with H3K9me2 (Figs. 2F, 3B). Previous work (the LET-418(A) data) revealed that LET-418, HPL-2, LIN-13, LIN-61, and MET-2 were all associated with H3K9me2 and repetitive elements whereas LET-418 exhibited generally lower enrichment on repetitive sequences than the other factors (Alicia N McMurchy, et al. 2017 Elife). We speculated that LET-418 may recognize both H3K4me3 and H3K9me3. The discrepancy might be caused by

different antibodies used (ab290 vs. Q3861) or worm culture conditions (OP50 vs. HB101 *E. coli*).

CEC-7 GFP(A) suggested that CEC-7 may associate with H3K4me3 (Figs. 2F, S9B). Although CEC-7 SDQ5421(A) revealed weak association with H3K4me3 at chromosome centers, CEC-7 SDQ5413(A) and CEC-7 SDQ5421(A) showed that CEC-7 may mainly associate with H3K36me3 and H3K79me3 (Figs. 2F, S9B, 3A-B). The CEC-7 SDQ5413(A) and CEC-7 SDQ5421(A) datasets were downloaded from modENCODE project, using antibody SDQ5413 and SDQ5421, respectively. The reason of discrepancy is unclear.

The discussion section was revised to include the information (lines 691-713).

In the revised Fig. 1E, peaks on LG III and LG X were shown. Peaks on other autosomes were exhibited in the new Figs. S7A-B.

• Regarding peak calling please provide an overview of how many peaks were called and the peak sizes. In addition please provide representative genome tracks showing examples of peaks with the aligned data. Especially heterochromatic proteins often show a broad distribution and for example peak calling can be very inaccurate for example for histone marks like H3K9me2/me3 because of that. Can the authors explain why they didn't choose the broad peak function integrated in MACS2?

RESPONSE: Thanks for the comments.

All chromodomain protein ChIP-seq peaks were called using MACS2 version 2.1.1 with subcommand callpeak used with parameters (-g ce -B -f BAM -q 0.01 -m 4 50). Size of most narrow peaks of chromodomain proteins was lower than 500bp (more than 60%) besides MRG-1(L4) (~33.48%). For MRG-1(L4), ~36.50% peaks were larger than 1kb. An overview of peak numbers and peak sizes was provided in Table S4. Representative genome tracks showing examples of peaks were included (new Figs. 6A-D).

We also called peaks using MACS2 version 2.1.1 with subcommand callpeak used with parameters (-g ce -B -f BAM -broad). We identified more broad peaks than narrow peaks for each chromodomain protein. Percentage of peaks >1kb was nearly equal to peaks <500bp (~30%). More than 90% narrow peaks were overlapped with broad peaks for each chromodomain proteins. In this work, we plotted heatmaps showing ChIP-seq signals near the center of each peak and analyzed correlation of pairs of samples (Figs. 2B-E, S8D-G, S9A, 3A, 5G and new Figs. S10, S11). Calling peaks using MACS2 with the default narrow peak function has an advantage in accuracy of signal intensity quantification across the peaks. Therefore, we used narrow peak calling for downstream analysis.

The methods section was revised (lines 792-807).

- *Page 8-9 (Line 238-247) please provide correlation coefficient for every correlation mentioned.*

RESPONSE: To evaluate the colocalization of chromodomain proteins with histone modifications identified by ChIP-seq assays on chromosome arms and centers, we performed scatter plots comparing log₂ enrichments of indicated histone marks versus chromodomain proteins mapped by ChIP-seq. A two-sided *t*-test was performed. $P < 2.2 \times 10^{-16}$ unless otherwise noted. Pearson correlation coefficient (*r*) was labeled on the plot of each pair of factors. $r \geq 0.300$ was colored by red and $r \leq -0.300$ was colored by green (new Figs. S10 and S11). The figures legends of S10 and S11 were revised accordingly (lines 1428-1442).

- *For Fig. 2a-d and Fig S2B-E and 3A: While the heatmaps are a information rich way of presenting the data, they are hard to translate into numbers and quantitative statements of enrichments. The additional provided correlation coefficients are in some way helpful, however for example UAD-2, CHD-7, SET-31 appear to cover the mutual exclusive H3K4me3 marked regions and H3K27me3 / H3K9me2/me3 regions. When analyzed together this weakens a overall correlation coefficient, despite clear overlap. It is therefore important to separate these and provide numbers (% of chromodomain peaks cover a certain PTM and vice versa).*

RESPONSE: Thanks for the suggestions.

To determine percentages of chromodomain peaks cover a certain PTM and vice versa, we used the IntervalStats software package. The method compared each single peak region from a ‘query’ experiment to the set of peak regions in a ‘reference’ experiment. P-values representative of the significance of query peak proximity to the reference peak were provided. Pairs of peaks were considered significantly overlapped with $P < 0.05$. Both chromodomain proteins and histone modifications were used as queries and references (new Figs. 2A, S8A-C). Percentage of overlap was listed in Table S7. We used the percentage of PTM covered peaks of each chromodomain protein for “graded histone occupied” evaluation. For many chromodomain proteins, PTM covered more than 30% of their peaks. However, for PTMs, chromodomain proteins rarely covered more than 30% (new Figs. S8B-C, Table S7). We speculated that for a given PTM, there are a number of additional readers besides chromodomain proteins, such as Tudor and MBT *etc.* may function redundantly to recognize the PTM.

The methods section was revised accordingly (lines 886-898).

- *One strategy could be: From Figure 2f I conclude that the majority of peaks cover a gene and a number of peaks cover intergenic sequences and repeats. In addition to the Fig.2a-d it would be informative to provide either heatmaps, or for the sake of clarity average coverage plots showing*

the coverage of the different chromodomain proteins at actively transcribed and silent genes (based on RNA expression, and Histone PTMs). For example centered around the transcription start site. There is no need to complicate things with a separation between “arms and center” as these are super broad regions and by no means a substitute for the proper correlation with histone PTMs. The analogous plots could be made with repetitive elements (e.g. split between DNA transposons, RNA transposons and simple repeats/satellite sequences). This would provide more information about how the chromodomain protein cover genes and or promoter and their relationship to the histone PTMs. In addition the authors should provide the % of chromodomain peaks cover a certain PTM and vice versa, as mentioned earlier.

RESPONSE: Thanks for the suggestions.

To identify actively transcribed genes and silent genes, we analyzed 2 repeats of mRNA-seq datasets of adult N2 worms. The average of Fragments Per Kilobase of exon model per Million mapped fragments (FPKM) of each gene was calculated. Actively transcribed genes were defined by “FPKM \geq 1” and being occupied by any of euchromatin marks (H3K4me3, H3K36me3, or H3K79me3). Silent genes are defined by “FPKM $<$ 1” and being occupied by any of heterochromatin marks (H3K9me2, H3K9me3, or H3K27me3).

In the new Figs. S13A-B, we plotted profile plots and heatmaps showing chromodomain protein ChIP-seq signals around transcription start sites (TSS) and transcript end sites (TES) of actively transcribed and silent targets. In the new Figs. S16A-C, we plotted profile plots and heatmaps showing chromodomain protein ChIP-seq signals at DNA transposons, Retrotransposons and other repetitive elements.

The results section (lines 303-305, line 338) and methods section (lines 913-920) were revised accordingly.

• Page 9: The authors conclude here that “Most chromodomain proteins were associated with both heterochromatin and euchromatin”, however ChIP-seq experiments were performed in whole animals and it is therefore likely that especially at genes that are differentially expressed in different tissues the authors look at a mixture of repressed and active genes and it is not clear whether the signal from the Chromodomain proteins reflects the binding to one or the other. As informative as their datasets are, this is a serious limitation and prohibits in my eyes conclusions that go beyond “protein x is enriched at chromatin state Y”.

RESPONSE: Thanks for the suggestions. We completely agree with the reviewer about the flaws of bulk ChIP-seq experiments.

The text was revised as “Most chromodomain proteins were enriched in both euchromatin and heterochromatin, yet the result may only reflect chromodomain protein occupancies and chromatin states of bulk tissues” (lines 297-299).

• *Page 10 Line 272: “UAD-2 was highly enriched in piRNA genes” Indeed more UAD-2 peaks appear to cover piRNA genes compared to the other chromodomain proteins. However from this graph it is not clear whether UAD-2 is enriched at piRNA genes compared to their occurrence in the genome (given the lower number of piRNA genes compared to coding genes that is very likely and the plot likely).*

RESPONSE: Thanks for the comments.

In the new Fig. S14B, we quantified UAD-2 ChIP-seq signals on piRNA and protein coding gene targets, respectively. UAD-2 ChIP-seq signals on piRNA targets were significantly higher than those on protein coding genes. Besides, UAD-2 localized to piRNA foci and was required for piRNA production (Huang, 2021 PNAS). Taken together, these data suggested that UAD-2 was highly enriched in piRNA genes.

The text was revised to include the information (lines 318-321).

• *Page 10 Line 279-280: “UAD-2 was not enriched in any specific GO (Gene Ontology) terms, which is consistent with its preferential binding to piRNA genes” This sentence does not make sense to me. According to Figure S4a the majority of UAD-2 peaks cover protein coding genes. Therefore, it is only a preferential binding to piRNA loci when compared to the other chromodomain proteins.*

RESPONSE: Thanks for the comments.

The text has been revised as “UAD-2 was not enriched in any specific GO (Gene Ontology) terms” (lines 327-328).

• *In addition to the correlation analysis and the coverage of the individual chromodomain proteins at genes and repeats it would be informative to plot the overlap of the chromodomain proteins at genes/repeats and then ask what is the chromatin state of the genes/repeats that show a certain combination of chromodomain proteins.*

RESPONSE: Thanks for the suggestions.

In the new Figs. S12A-B, we plotted the overlap of the chromodomain proteins at genes/repeats and compared the chromatin state of the targets showing a certain combination of chromodomain proteins. The result was largely consistent with the “graded histone modification occupied” analysis (Figs. 2F and S9B). For example, the “graded histone modification occupied” analysis showed CEC-5 GFP(A), CEC-8 GFP(E), HPL-2(L3), and HPL-2(A) targets were all associated with H3K9me2. A combination of CEC-5 GFP(A), CEC-8 GFP(E), HPL-2(L3), HPL-2(A) targets (both genes and repeats) were occupied by H3K9me2 as well. Figs. 2F and S9B showed that CHD-1 GFP(A), CHD-3 GFP(A), and CHD-3 SDQ3907(A) targets were all associated with H3K4me3. A combination of CHD-1 GFP(A), CHD-3 GFP(A), and CHD-3

SDQ3907(A) targets (both genes and repeats) were also occupied by H3K4me3, although weaker H3K36me3 and H3K79me3 signals were also detected.

The text was revised to include the information (lines 284-295).

• *Page 14 (Line 411-424): Please discuss that because of the lack of a spike in control, one can not say whether the appearance of novel CEC-5 peaks in a met-2 mutant is due to an increased binding of CEC-5 to these regions, or whether there is a general loss of chromatin association of CEC-5 and the signal observed in the ChIP-seq is noise amplified due to PCR and normalization of the data.*

RESPONSE: Thanks for the comments.

We agree with the reviewer that because of the lack of a spike in control, it is unclear whether the appearance of novel CEC-5 peaks in *met-2* mutant is due to an increased binding of CEC-5 to these regions, or whether there is a general loss of chromatin association of CEC-5, and the signal observed in the ChIP-seq is noise amplified due to PCR and normalization.

We revised the results section to include the possibility (lines 494-498).

3) The link between CEC-5 and the nucleolus is interesting, especially in the context of the aging phenotype, however it remains weak. It would be strengthened if the authors could quantify rRNA expression in WT, met-2 and cec-5 mutants. In flies loss of Su(var)3-9 results in the fragmentation of nucleoli and rDNA instability (Peng et al., 2007 and 2009). The authors could check if this is also the case for CEC-5 mutant embryos (e.g. using the FIB-1::GFP strain).

RESPONSE: Thanks for the suggestions.

To explore the link between CEC-5 and the nucleolus, we quantified rRNA expression levels in *WT*, *met-2* and *cec-5* mutants by quantitative real time PCR. However, we failed to detect pronounced alterations on mature rRNA (18S rRNA, 5.8S rRNA, and 26S rRNA) levels (new Fig S20D). Moderate increase of pre-rRNA levels was detected in *met-2* and *cec-5* mutants (new Fig. S20E). The nucleolus morphology in either *met-2* or *cec-5* mutants were not noticeably changed either (new Fig. S20F). The role of CEC-5 in nucleolus are still under detailed investigation. Hopefully, we can understand how and why of CEC-5 in nucleolus in the near future.

The results (lines 556-561) sections were revised accordingly.

Minor points:

I would ask the authors to change the colors of the c elegans proteins in Fig. 1a to make it more noticeable. Also please distinguish human and C. elegans proteins only by color and remove the

individual text to make it more accessible.

RESPONSE: We have changed the colors and removed the individual text as suggested (Fig. 1A).

Page 11 Line 309: please cite Garrigues et al., 2015 for HPL-2; Padeken et al., 2019 and 2021 for MET-2 and LIN-61 and Koester-Eiserfunke et al., 2011 for LIN-61

RESPONSE: Sorry of for the missing. These references have been cited in the revision (lines 356-358).

Page 15 Line 439-441: “Surprisingly, CEC-5 bound both unmethylated (me0) and methylated (me1/2/3) H3K9 peptides, although CEC5 CD showed weaker binding affinity to unmethylated (me0) H3K9 peptide (Figs.6C, 6D).” please add a statement that this reflects the binding of only the isolated chromodomain under these invitro conditions. In my eyes the low affinity to H3K9me0 in this assay plus the actual clear invivo dependency on H3K9me1/me2 for binding makes it far more likely that CEC-5 is indeed binding H3K9 when methylated.

RESPONSE: Thanks for the comments.

We agree with the reviewer that the low affinity to H3K9me0 in the in vitro binding assay plus the actual in vivo dependency on H3K9me1/me2 for binding strongly suggested that CEC-5 binds methylated H3K9.

We have revised the text as: “Although CEC-5 chromodomain exhibited low affinity to H3K9me0 in the in vitro binding assay, the in vivo dependency on *met-2* for binding suggested that CEC-5 protein binds methylated H3K9. In addition, the binding of CEC-5 to chromosomes was likely independent of SET-25 in vivo, by both ChIP-seq and subcellular localization. MET-2 is essential for ~80% of all the H3K9me3 catalyzed by SET-25 and in the absence of SET-25 these regions retain H3K9me2 (Padeken et al., 2019). These data suggested that CEC-5 recognizes H3K9me1/2, but not H3K9me3, in vivo” (lines 517-523).

Page 15 Line 439-441: “Since the binding of CEC-5 to chromosomes depends on H3K9me1/2 but not H3K9me3 in vivo, we speculated that other amino acid sequences of CEC-5 may modulate the binding specificity of the CEC-5 chromodomain” Please note that H3K9me2 is sufficient for CEC-5 binding. This doesn’t mean that it does not H3K9me3. The H3K9me1/me2 provided by MET-2 is essential for ~80% of all the H3K9me3 catalyzed by SET-25 and in the absence of SET-25 these regions retain H3K9me2. (Padeken et al., 2019).

RESPONSE: Thanks for the comments. We kindly disagree with the reviewer.

In *set-25* mutants, most if not all H3K9me3 was depleted, yet *set-25* does not affect the subcellular localization and ChIP-seq of CEC-5. Therefore, it is very unlikely that CEC-5 recognizes H3K9me3 in vivo.

We have revised the text as: “Although CEC-5 chromodomain exhibited low affinity to H3K9me0 in the in vitro binding assay, the in vivo dependency on *met-2* for binding suggested that CEC-5 protein binds methylated H3K9. In addition, the binding of CEC-5 to chromosomes was likely independent of SET-25 in vivo, by both ChIP-seq and subcellular localization. MET-2 is essential for ~80% of all the H3K9me3 catalyzed by SET-25 and in the absence of SET-25 these regions retain H3K9me2 (Padeken et al., 2019). These data suggested that CEC-5 recognizes H3K9me1/2, but not H3K9me3, in vivo” (lines 517-523).

In the same context please change the title. Given the invitro data and the correct interpretation of the genetic experiment one can not conclude that CEC-5 is a “H3K9me1/2 reader” instead it is a reader of H3K9me1/me2/me3 (or H3K9me).

RESPONSE: As described above, the association of CEC-5 is independent of SET-25 strongly suggested that CEC-5 is not a H3K9me3 reader. Therefore, we prefer to keep the title as H3K9me1/2 reader.

Page 17 Line 496-497: “We identified a point mutation in the CEC-5 chromodomain critical for its recognition of methyl-lysine of histone 3” Please rephrase: In the data presented in Figure 7 the authors showed that the R124C mutation results in a loss of proper CEC-5 localization. This does not show whether it is due to an inability to bind H3K9me.

RESPONSE: Thanks for the suggestions. However, we kindly disagree with the reviewer.

In the new Fig. 7F, we showed that CEC-5(R124C) reduced the binding to H3K9me associated heterochromatin in vivo by ChIP-seq assay.

REVIEWERS' COMMENTS

Reviewer #1 (Remarks to the Author):

The authors have addressed the points raised in the review.

Reviewer #2 (Remarks to the Author):

In the revised version of the manuscript "Systematic characterization of chromodomain proteins reveals an H3K9me1/2 reader regulating aging in *C. elegans*" Hou, Xu, Zhu and colleagues addressed my major concerns.

My only remaining issue is the interpretation of CEC-5 as a reader of H3K9me1/me2, but not H3K9me3.

I have 2 problems with this interpretation:

2nd: The new Figure 5B-F appear to show that in a met-2 set-25 double mutant CEC-5 remains bound to the majority of CEC-5 peaks found in wt animals. Taken in isolation this would argue that CEC-5 might also bind independently of H3K9me. However, the microscopy (Figure 4F) and the genome tracks presented in Figure 5A strongly suggest that in the absence of met-2 set-25 CEC-5 binding is either unspecific or absent and the apparent signal in Figure 5B-F a artifact of this specific way to represent the data. I would ask the authors to discuss these possibilities in the results.

1st: Strictly speaking the authors have shown that CEC-5 binds H3K9me1/me2/me3 in vitro and that in the absence of H3K9me3 (set-25 mutant) CEC-5 is still able to bind chromatin in vivo. This means that H3K9me3 is not essential for CEC-5 chromatin binding. However, it does not mean that CEC-5 only binds H3K9me1/me2 and not H3K9me3. Logically speaking: If also in vivo CEC-5 would bind all three methylation states, removing H3K9me3, but retaining H3K9me1/me2 would equally explain why there is no change in CEC-5 binding in a set-25 mutant. In fact in a set-25 mutant H3K9me2 levels are increased, because MET-2 is still able to catalyse H3K9me1/me2 at 80% of the heterochromatin domains which are marked by H3K9me3 in wild-type. If CEC-5 would be a strict H3K9me1/me2, but not H3K9me3 reader I would expect to see an equal increase of CEC-5 binding in a set-25 mutant in regions marked by H3K9me3 in wild type animals.

I appreciate that this is a detail and does not change the overall conclusion of the manuscript. However with the current data the authors simply can not claim that CEC-5 does not bind H3K9me3. I would suggest that the authors simply change their statement "...our data suggested that CEC-5 recognizes H3K9me1/2, but not H3K9me3, in vivo." into: "our data suggested that CEC-5 recognizes H3K9me1/2 in vivo." (Line: 522-523).

REVIEWERS' COMMENTS

Reviewer #1 (Remarks to the Author):

The authors have addressed the points raised in the review.

RESPONSE: Thanks very much for the comments.

Reviewer #2 (Remarks to the Author):

In the revised version of the manuscript “Systematic characterization of chromodomain proteins reveals an H3K9me1/2 reader regulating aging in C. elegans” Hou, Xu, Zhu and colleagues addressed my major concerns.

My only remaining issue is the interpretation of CEC-5 as a reader of H3K9me1/me2, but not H3K9me3.

I have 2 problems with this interpretation:

2nd: The new Figure 5B-F appear to show that in a met-2 set-25 double mutant CEC-5 remains bound to the majority of CEC-5 peaks found in wt animals. Taken in isolation this would argue that CEC-5 might also bind independently of H3K9me. However, the microscopy (Figure 4F) and the genome tracks presented in Figure 5A strongly suggest that in the absence of met-2 set-25 CEC-5 binding is either unspecific or absent and the apparent signal in Figure 5B-F a artifact of this specific way to represent the data. I would ask the authors to discuss these possibilities in the results.

RESPONSE: Thanks very much for the suggestions. We agree with the reviewer.

We have revised the results section as suggested by the reviewer. (Lines 479-486)

“These data suggested that MET-2 and H3K9me1/2 were required for the proper association of CEC-5 with chromatin. Alternatively, the fact that in *met-2* single mutant and *met-2;set-25* double mutant, CEC-5 remained bound to the majority of CEC-5 peaks (Figs. 5b-f) found in wild-type animals may suggest that CEC-5 might also bind to chromatin independently of H3K9me status. Since the microscopy experiments (Fig. 4f) and genome tracks (Fig. 5a) suggested that *met-2* was required for CEC-5 binding, the ChIP signal observed in heatmaps (Figs. 5b-f) could be artifacts from the way of representing data, which requires further investigation.

1st: Strictly speaking the authors have shown that CEC-5 binds H3K9me1/me2/me3 in vitro and that in the absence of H3K9me3 (set-25 mutant) CEC-5 is still able to bind chromatin in vivo. This means that H3K9me3 is not essential for CEC-5 chromatin binding. However, it does not mean that CEC-5 only binds H3K9me1/me2 and not H3K9me3. Logically speaking: If also in vivo CEC-5 would bind all three methylation states, removing H3K9me3, but retaining H3K9me1/me2 would equally explain why there is no change in CEC-5 binding in a set-25 mutant. In fact in a set-25 mutant H3K9me2 levels are increased, because MET-2 is still able to catalyse H3K9me1/me2 at

80% of the heterochromatin domains which are marked by H3K9me3 in wild-type. If CEC-5 would be a strict H3K9me1/me2, but not H3K9me3 reader I would expect to see a equal increase of CEC-5 binding in a set-25 mutant in regions marked by H3K9me3 in wild type animals.

I appreciate that this is a detail and does not change the overall conclusion of the manuscript. However with the current data the authors simply can not claim that CEC-5 does not bind H3K9me3. I would suggest that the authors simply change their statement "...our data suggested that CEC-5 recognizes H3K9me1/2, but not H3K9me3, in vivo." into: "our data suggested that CEC-5 recognizes H3K9me1/2 in vivo." (Line: 522-523).

RESPONSE: Thanks for the suggestions.

We agree with the reviewer and have revised the text as suggested (lines 519-520):

"Our data suggested that CEC-5 recognizes H3K9me1/2 in vivo".